# Efficient coding, channel capacity, and the emergence of retinal mosaics

**Na Young Jun**
Department of Neurobiology
Duke University
Durham, NC 27710
nayoung.jun@duke.edu

**Greg D. Field**
Department of Neurobiology
Duke University
Durham, NC 27710
field@neuro.duke.edu

**John M. Pearson**
Department of Biostatistics & Bioinformatics
Department of Neurobiology
Department of Electrical and Computer Engineering
Duke University
Durham, NC 27710
john.pearson@duke.edu

## Abstract

Among the most striking features of retinal organization is the grouping of its output neurons, the retinal ganglion cells (RGCs), into a diversity of functional types. Each of these types exhibits a mosaic-like organization of receptive fields (RFs) that tiles the retina and visual space. Previous work has shown that many features of RGC organization, including the existence of ON and OFF cell types, the structure of spatial RFs, and their relative arrangement, can be predicted on the basis of efficient coding theory. This theory posits that the nervous system is organized to maximize information in its encoding of stimuli while minimizing metabolic costs. Here, we use efficient coding theory to present a comprehensive account of mosaic organization in the case of natural videos as the retinal channel capacity—the number of simulated RGCs available for encoding—is varied. We show that mosaic density increases with channel capacity up to a series of critical points at which, surprisingly, new cell types emerge. Each successive cell type focuses on increasingly high temporal frequencies and integrates signals over larger spatial areas. In addition, we show theoretically and in simulation that a transition from mosaic alignment to anti-alignment across pairs of cell types is observed with increasing output noise and decreasing input noise. Together, these results offer a unified perspective on the relationship between retinal mosaics, efficient coding, and channel capacity that can help to explain the stunning functional diversity of retinal cell types.

## 1 Introduction

The retina is one of the most intensely studied neural circuits, yet we still lack a computational understanding of its organization in relation to its function. At a structural level, the retina forms a three-layer circuit, with its primary feedforward pathway consisting of photoreceptors to bipolar cells to retinal ganglion cells (RGCs), the axons of which form the optic nerve [1]. RGCs can be divided into 30-50 functionally distinct cell types (depending on species) with each cell responsive to a localized area of visual space (its receptive field (RF)), and the collection of RFs for each type tiling space to form a "mosaic" [2, 3, 4, 5]. Each mosaic represents the extraction of a specific type

36th Conference on Neural Information Processing Systems (NeurIPS 2022).

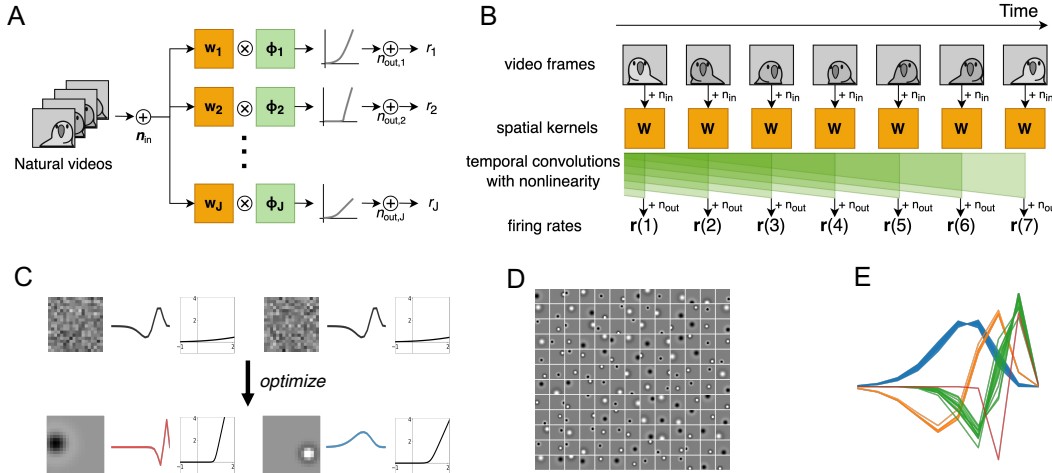

Figure 1: ON and OFF RF mosaics and their temporal kernels are predicted by efficient coding of natural videos. **(A)** Frames of natural videos $\mathbf{x}(t)$ plus input noise $\mathbf{n}_{in}$ are linearly filtered with the spatial kernels $\mathbf{w}_j$ and then passed through one-dimensional temporal convolutions $\phi_j$ followed by a nonlinearity, resulting firing rates $r_j(t)$ for each of $J$ RGCs. **(B)** The same calculations shown along the time axis, visualizing the temporal convolutions. **(C)** Examples of initial and optimized spatial filters, temporal filters, and nonlinearities: **(left)** *fast OFF* kernel, **(right)** *slow ON* kernel. **(D)** Unconstrained spatial filters ($J = 160$) learned center-surround shapes, about half of which are ON RFs. **(E)** Temporal filters ($J = 160$) using the parameterization (6) converged to four distinct clusters.

of information across the visual scene by a particular cell type, with different mosaics responding to light increments or decrements (ON and OFF cells), high or low spatial and temporal frequencies, color, motion, and a host of other features. While much is known about the response properties of each RGC type, the computational principles that drive RGC diversity remain unclear.

Efficient coding theory has proven one of the most powerful ideas for understanding retinal organization and sensory processing. Efficient coding posits that the nervous system attempts to encode sensory input by minimizing redundancy subject to biological costs and constraints [6, 7]. As more commonly formulated, it seeks to maximize the mutual information between sensory data and neural representations, with the most common cost in the retinal case being the energetic cost of action potentials transmitted by the RGCs. Despite its simplicity, this principle has proven useful, predicting the center-surround structure of RFs [8], the frequency response profile of contrast sensitivity [9], the structure of retinal mosaics [10, 11], the role of nonlinear rectification [12], different spatiotemporal kernels [13], and inter-mosaic arrangements [14, 15].

While previous studies have largely focused on either spatial or temporal aspects of efficient coding, we optimize an efficient coding model of retinal processing in both space and time to *natural videos* [16]. We systematically varied the number of cells available to the system and found that larger numbers of available cells led to more cell types. Each of these functionally distinct types formed its own mosaic of RFs that tiled space. We show that when and how new cell types emerge and form mosaics is the result of tradeoffs between power constraints and the benefits of specialized encoding that shift as more cells are available to the system. We show that cell types begin by capturing low-frequency temporal information and capture increasingly higher-frequency temporal information over larger spatial RFs as new cell types form. Finally, we investigated the relative arrangement of these mosaics and their dependence on noise. We show that mosaic pairs can be aligned or anti-aligned depending on input and output noise in the system [14]. Together, these results demonstrate for the first time how efficient coding principles can explain, even predict, the formation of cell types and which types are most informative when channel capacity is limited.

## 2 Model

The model we develop is an extension of [14], a retinal model for efficient coding of natural images, which is based on a mutual information maximization objective proposed in [10]. The retinal model takes $D$-pixel patches of natural images $\mathbf{x} \in \mathbb{R}^D$ corrupted by input noise $\mathbf{n}_{\mathrm{in}} \sim \mathcal{N}(0, \mathbf{C}_{n_{\mathrm{in}}})$, filters these with unit-norm linear kernels $\{\mathbf{w}_j \in \mathbb{R}^D \mid \|\mathbf{w}_j\| = 1\}_{j=1,\cdots,J}$ representing $J$ RGCs, and then feeds the resulting signals $y_j = \mathbf{w}_j^\top (\mathbf{x} + \mathbf{n}_{\mathrm{in}})$ through softplus nonlinearities $\eta(y) = \log\left(1 + e^{\beta y}\right)/\beta$ (we used $\beta = 0.25$) with gain $\gamma_j$ and threshold $\theta_j$. Finally, these signals are further corrupted by additive output noise $\mathbf{n}_{\mathrm{out}} \sim \mathcal{N}(0, \mathbf{C}_{n_{\mathrm{out}}})$, to produce firing rates $r_j$:

$$r_j = \gamma_j \cdot \eta(y_j - \theta_j) + n_{\mathrm{out},j}. \tag{1}$$

The model learns parameters $\mathbf{w}_j$, $\gamma_j$, and $\theta_j$ to maximize the mutual information between the inputs $\mathbf{x}$ and the outputs $\mathbf{r}$, under a mean firing rate constraint [10, 14]:

$$\text{maximize} \quad \log \frac{\det\left(\mathbf{G}\mathbf{W}^\top\left(\mathbf{C_x} + \mathbf{C}_{n_{\mathrm{in}}}\right)\mathbf{W}\mathbf{G} + \mathbf{C}_{n_{\mathrm{out}}}\right)}{\det\left(\mathbf{G}\mathbf{W}^\top \mathbf{C}_{n_{\mathrm{in}}}\mathbf{W}\mathbf{G} + \mathbf{C}_{n_{\mathrm{out}}}\right)} \tag{2}$$

$$\text{subject to} \quad \mathbb{E}[r_j] = 1. \tag{3}$$

Here $\mathbf{C_x}$ is the covariance matrix of the input distribution, $\mathbf{W} \in \mathbb{R}^{D \times J}$ contains the filters $\mathbf{w}_j$ as its columns, the gain matrix $\mathbf{G} = \mathrm{diag}\left(\gamma_j \frac{d\eta}{dy}\big|_{y_j - \theta_j}\right)$, and the noise covariances are $\mathbf{C}_{n_{\mathrm{in}}} = \sigma_{\mathrm{in}}^2 \mathbb{1}_{D \times D}$ and $\mathbf{C}_{n_{\mathrm{out}}} = \sigma_{\mathrm{out}}^2 \mathbb{1}_{J \times J}$. This objective is equivalent to the formulation in [10], which assumes normally distributed inputs and locally linear responses in order to approximate the mutual information in a closed form.

Here, we extend this model to time-varying inputs $\mathbf{x}(t) \in \mathbb{R}^D$ representing natural videos (Figure 1A-B), which are convolved with linear spatiotemporal kernels $\{\mathbf{w}_j(t)\}_{j=1,\cdots,J}$:

$$y_j(t) = \mathbf{w}_j^\top(t) * \mathbf{x}(t) = \int_{-\infty}^{\infty} \mathbf{w}_j(\tau)^\top \mathbf{x}(t - \tau) d\tau. \tag{4}$$

We additionally assume that the convolutional kernels are separable in time and space:

$$\mathbf{w}_j(t) = \phi_j(t)\mathbf{w}_j, \qquad \|\mathbf{w}_j\| = 1, \quad \phi_j(t) \in \mathbb{R}, \quad \int_{-\infty}^{\infty} \phi(t)^2 dt = 1, \tag{5}$$

and the temporal kernels are unit-norm impulse responses taking the following parametric form:

$$\phi_j(t) \propto \begin{cases} \alpha_j t^n e^{-t/\tau_j} - \alpha_j' t^n e^{-t/\tau_j'} & \text{if delay } t \geq 0 \\ 0 & \text{otherwise} \end{cases}, \tag{6}$$

where $\alpha_j, \alpha_j', \tau_j > 0, \tau_j' > 0$ are learnable parameters, and $n \in \mathbb{N}$ is fixed. Previous work assumed an unconstrained form for these filters, adding zero-padding before and after the model's image inputs to produce the characteristic shape of the temporal filters in primate midget and parasol cells [13], but this zero-padding represents a biologically implausible constraint, and the results fail to correctly reproduce the observed delay in retinal responses [17, 18, 19]. Rather, optimizing (2) with unconstrained temporal filters produces a filter bank uniformly tiling time (Supplementary Figure 4).

By contrast, (6) is motivated by the arguments of [20], which showed that the optimal minimum-phase temporal filters of retinal bipolar cells, the inputs to the RGCs, take the form

$$\phi(t > 0) \propto e^{-t/\tau}[\sin \omega t - \omega t \cos \omega t] \approx e^{-t/\tau}\frac{(\omega t)^3}{3} \tag{7}$$

when $\omega\tau \ll 1$. Thus, we model RGC temporal filters as a linear combination of these forms. In practice, we take only two filters and use $n = 6$ rather than $n = 3$, since these have been shown to perform well in capturing observed retinal responses [19]. The results produced by more filters or different exponents are qualitatively unchanged (Supplementary Figure 7). For training on video data, we use discrete temporal filters and convolutions with $\sum_{t=0}^{T-1} \phi_j[t]^2 = 1$. Finally, while unconstrained spatial kernels $\mathbf{w}_j$ converge to characteristic center-surround shapes under optimization of (2) (Figure 1C), for computational efficiency and stability, we parameterized these filters using a radially-symmetric difference of Gaussians

$$w_j(r) \propto e^{-a_j r^2} - c_j e^{-b_j r^2}, \qquad b_j > a_j > 0, \quad 0 < c_j < 1, \tag{8}$$

where $r$ measures the spatial distance to the center of the RF, and the parameters $a_j$, $b_j$, $c_j$ that determine the center location and spatial kernel shape are potentially different for each RGC $j$. The result of optimizing (2) using these forms is a set of spatial and temporal kernels (Figure 1D-E) that replicate experimentally-observed shapes and spatial RF tiling.

## 3 Efficient coding as a function of channel capacity: linear theory

Before presenting results from our numerical experiments optimizing the model (2, 3), we begin by deriving intuitions about its behavior by studing the case of *linear* filters analytically. That is, we assume a single gain $\gamma$ for all cells, no bias ($\theta = 0$), and a linear transfer function $\eta(y) = y$. As we will see, this linear analysis correctly predicts the same types of mosaic formation and filling observed in the full nonlinear model. Here, we sketch the main results, deferring full details to Appendix A.

### 3.1 Linear model in the infinite retina limit

For analytical simplicity, we begin by assuming an infinite retina on which RFs form mosaics described by a regular lattice. Under these conditions, we can write the log determinants in (2) as integrals and optimize over the *unnormalized* filter $v \equiv \gamma w$ subject to a power constraint:

$$\max_v \ \int_{G_0} \frac{d^2\mathbf{k}}{(2\pi)^2} \left[ \log \frac{\sum_{\mathbf{g}\in G} |v(\mathbf{k}+\mathbf{g})|^2 (C_x(\mathbf{k}+\mathbf{g}) + \sigma_{\text{in}}^2) + \sigma_{\text{out}}^2}{\sum_{\mathbf{g}\in G} |v(\mathbf{k}+\mathbf{g})|^2 \sigma_{\text{in}}^2 + \sigma_{\text{out}}^2} - \nu \sum_{\mathbf{g}\in G} |v(\mathbf{k}+\mathbf{g})|^2 (C_x(\mathbf{k}+\mathbf{g}) + \sigma_{\text{in}}^2) \right], \quad (9)$$

where $C_x(\mathbf{k})$ is the Fourier transform of the stationary image covariance $C_x(\mathbf{z} - \mathbf{z}')$, the integral is over all frequencies $\mathbf{k} \in G_0$ *unique up to aliasing* caused by the spatial regularity of the mosaic, and the sums over $\mathbf{g}$ account for aliased frequencies (Appendix A.1). In [8], the range $[-\pi, \pi]$ is used for the integral, corresponding to a one-dimensional lattice and units of mosaic spacing $\Delta z = 1$.

Now, solving the optimization in (9) results in a spatial kernel with the spectral form (Appendix A.2)

$$|v(k)|^2 = \frac{\sigma_{\text{out}}^2}{\sigma_{\text{in}}^2} \left[ \frac{1}{2} \frac{C_x(k)}{C_x(k) + \sigma_{\text{in}}^2} \left( \sqrt{1 + \frac{\sigma_{\text{in}}^2}{\sigma_{\text{out}}^2} \frac{4}{\nu \, C_x(k)}} + 1 \right) - 1 \right]_+, \quad k \in G_0, \quad (10)$$

where $k = \|\mathbf{k}\|$ and $\nu$ is chosen to enforce the constraint on total power. This is exactly the solution found in [8], linking it (in the linear case) to the model of [10, 11]. Note, however, that (10) is only nonzero within $G_0$, since RF spacing sets an upper limit on the passband of the resulting filters.

The generalization of this formulation to the spacetime case is straightforward. Given a spacetime stationary image spectrum $C_x(\mathbf{z} - \mathbf{z}', t - t')$ and radially-symmetric, causal filter $w(\mathbf{z}, t)$, the same infinite retina limit as above requires calculating determinants across both neurons $i, j$ and time points $t, t'$ of matrices with entries of the form

$$F_{ijtt'} = \int d\mathbf{z} d\mathbf{z}' d\tau d\tau' \gamma^2 w(\mathbf{z}_i - \mathbf{z}, t - \tau) C_x(\mathbf{z} - \mathbf{z}', \tau - \tau') w(\mathbf{z}_j - \mathbf{z}', t' - \tau')$$

$$= \int \frac{d^2\mathbf{k}}{(2\pi)^2} \frac{d\omega}{2\pi} e^{i\mathbf{k}\cdot(\mathbf{z}_i - \mathbf{z}_j) + i\omega(t - t')} |v(k, \omega)|^2 C_x(k, \omega). \quad (11)$$

Again, such matrices can be diagonalized in the Fourier basis, with the result that the optimal spacetime filter once again takes the form (10) with the substitutions $v(k) \to v(k, \omega)$, $C_x(k) \to C_x(k, \omega)$ (Appendix A.3). Figure 2A depicts the frequency response of this filter in $d = 1$ spatial dimensions, with corresponding spatial and temporal sections plotted in Figures 2B-C.

### 3.2 Multiple cell types and the effects of channel capacity

Up to this point, we have only considered a single type of filter $v(k, \omega)$, corresponding to a single cell type. However, multiple cell types might increase the coding efficiency of the entire retina if they specialize, devoting their limited energy budget to non-overlapping regions of frequency space. Indeed, optimal encoding in the multi-cell-type case selects filters $v$ and $v'$ that satisfy $v^*(k, \omega) v'(k, \omega) = 0$, corresponding encoding *independent* visual information (Appendix A.4).

This result naturally raises two questions: First, how many filter types are optimal? And second, how should a given budget of $J$ RGCs be allocated across multiple filter types? As detailed in Appendix

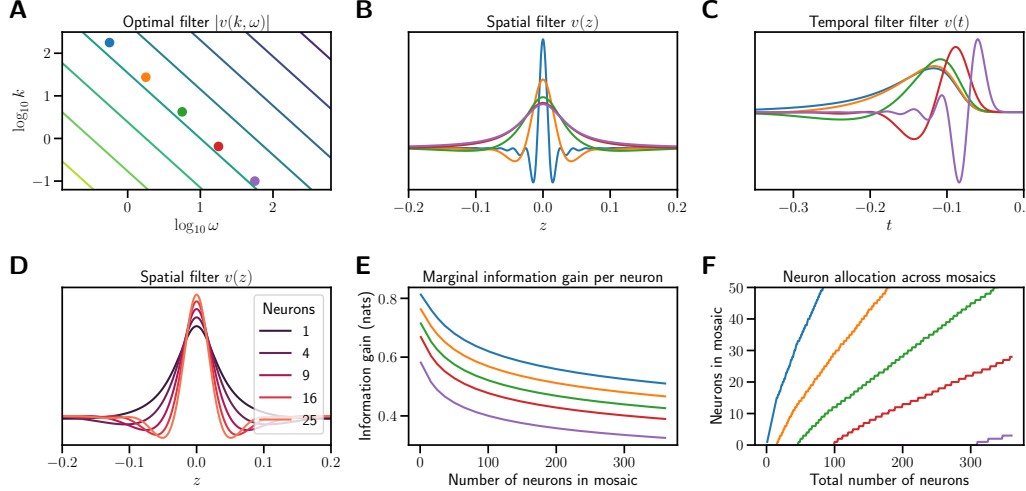

Figure 2: Optimal filters in the linear spacetime case. **(A)** Spectrum of the optimal linear spacetime filter in $d = 1$ spatial dimension. Contour lines indicate constant (log) power. **(B)** Spatial filters at representative temporal frequencies. Each filter represents a vertical section at the correspondingly colored dot in **A**. **(C)** Temporal filters at representative spatial frequencies. Each filter represents a horizontal section at the correspondingly colored dot in **A**. **(D)** Spatial filters at $\omega = 0$ for increasing numbers of RGCs $J$. The spatial extent of the center narrows with more RGCs added to the mosaic. **(E)** Gain in information per RGC added to each mosaic as a function of current RGC number $J$. New cell types begin when the marginal benefit of adding an RGC to an existing mosaic equals the benefit of adding the first RGC of a new cell type. Color indicates $\omega_0$, the temporal frequency of the narrowband filter. **(F)** Total number of RGCs in each mosaic as a function of total RGCs across all mosaics. As new cell types arise and form mosaics, new RGCs are allocated to existing mosaics at a decreasing rate. For both plots, $A = 100$, $\sigma_{\text{in}} = 0.4$, $\sigma_{\text{out}} = 1.25$, and $\log_{10} \omega_0 = 1.5, 1.52, 1.54, 1.56, 1.6$. Details of calculations in Appendix A.5.

A.5, we can proceed by analyzing the case of a finite retina in the Fourier domain, approximating the information encoded by a mosaic of $J$ RGCs with spatial filters given by (10) and nonoverlapping bandpass temporal filters that divide the available spectrum (e.g., Figure 2B, C). Following [21], we approximate the correlation spectrum of images by the factorized power law $C_x(k, \omega) \simeq \frac{A}{k^\alpha \omega^2}$ with $\alpha \approx 1.3$ and find that in this case, the optimal filter response exhibits two regimes as a function of spatial frequency (Supplementary Figure 1A): First, below $k_f = \left(A/\sigma_{\text{in}}^2 \omega^2\right)^{1/\alpha}$, the optimal filter is separable and log-linear, and the filtered image spectrum is white:

$$|v(k, \omega)|^2 \approx \frac{k^\alpha \omega^2}{A\nu}, \qquad |v(k, \omega)|^2 C_x(k, \omega) \approx \nu^{-1},$$

where $\nu$, the Lagrange multiplier in (9) that enforces the power constraint, scales as $1/P$ for small values of maximal power $P$ and $1/P^2$ for larger values (Supplementary Figure 1D). Second, for $k \gtrsim k_f$, the filter response decreases as $k^{-\alpha/2}$ until reaching its upper cutoff at $k_c = k_f/(\nu \sigma_{\text{out}}^2)^{1/\alpha}$, with the filtered image spectrum falling off at the same rate (Supplementary Figure 1B).

But what do these regimes have to do with mosaic formation? The link between the two is given by the fact that, for a finite retina with regularly spaced RFs, adding RGCs decreases the distance between RF centers and so increases the resolving power of the mosaic. That is, the maximal value of $k$ grows roughly as $k \sim \sqrt{J}$ in $d = 2$, such that larger numbers of RGCs capture more information at increasingly higher spatial frequencies (Supplementary Figure 1A). However, while information gain is roughly uniform in the whitening regime, it falls off sharply for $k \gtrsim k_f$ (Supplementary Figure 1C), suggesting the interpretation that the $k \lesssim k_f$ regime is a "mosaic filling" phase in which information accumulates almost linearly as RFs capture new locations in visual space, while the $k \gtrsim k_f$ regime constitutes a "compression phase" in which information gains are slower as RFs shrink to accommodate higher numbers (Figure 2D). Indeed, one can derive the scaling of total

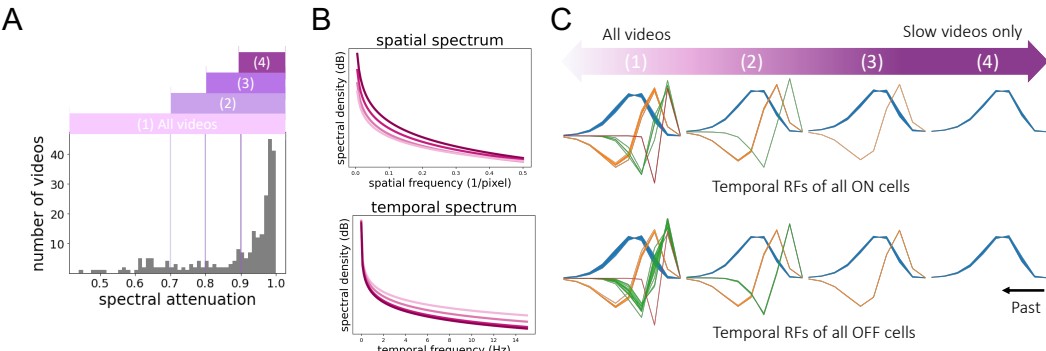

Figure 3: Statistics of natural videos affect learned RFs. **(A)** Histogram of spectral attenuation (fraction of power $< 3$ Hz) for each video clip from the Chicago Motion Database. A significant portion of the dataset exhibits predominaly low-frequency spectral content in time. Videos with spectral attenuation above 0.9, 0.8, and 0.7, are denoted (4), (3), and (2), respectively, while (1) refers to all videos in the dataset. **(B)** Spatial (top) and temporal (bottom) spectral density of the four subsets. **(C)** Temporal filters learned by training on each of the four subsets. Training on slow videos produced only smoothing kernels, while training on all videos produced a variety of temporal filters.

information as a function of $J$:

$$
\mathcal{I} \simeq \begin{cases} J\left[\log\left(1 + \frac{P_0}{\sigma_{\text{out}}^2}\right) - \frac{2P_0}{(\alpha+2)\sigma_{\text{out}}^2}\left(\frac{J}{J_f}\right)^{\frac{\alpha}{2}}\right] & k \lesssim k_f \\ (J - J_f)\left(\frac{J}{J_f}\right)^{-\frac{\alpha}{2}}\frac{2}{2-\alpha} & k \gtrsim k_f \end{cases},
\tag{12}
$$

where $P_0$ is the power budget per RGC and $J_f$ is the RGC number corresponding to $k = k_f$. Thus, mosaic filling exhibits diminishing marginal returns (Figure 2E), such that new cell types are favored when the marginal gain for growing mosaics with lower temporal frequency drops below the gain from initiating a new cell type specialized for higher temporal frequencies. Moreover, the difference between these gain curves implies that new RFs are not added to all mosaics at equal rates, but in proportion to their marginal information (Figure 2F). As we demonstrate in the next section, these features of cell type and mosaic formation continue to hold in the full nonlinear model in simulation.

## 4 Experiments

We analyzed the characteristics of the optimal spatiotemporal RFs obtained from the model (2, 3) trained on videos from the Chicago Motion Database [22]. Model parameters for spatial kernels, temporal kernels, and the nonlinearities were jointly optimized using Adam [23] to maximize (2) subject to the mean firing rate constraint (3) using the augmented Lagrangian method with the quadratic penalty $\rho = 1$ [24]. Further technical details of model training are in Appendix E. All model code and reproducible examples are available at `https://github.com/pearsonlab/efficientcoding`.

As previously noted, the power spectral density of natural videos can be well approximated by a product of spatial and temporal power-law densities, implying an anticorrelation between high spatial and temporal frequency content [21]. Supplementary Figure 5 shows the data spectrum of the videos in our experiments is also well-approximated by separable power-law fits. To examine the effect of these statistics on the learned RFs, we divided the dataset into four progressively smaller subsets by the proportion of their temporal spectral content below 3 Hz, their *spectral attenuation*. Using values of 70%, 80%, and 90% then yielded a progression of datasets ranging from most videos to only the slowest videos (Figure 3A, B). Indeed, when the model was trained on these progressively slower data subsets, it produced only temporal smoothing filters, whereas the same model trained on all videos produced a variety of "fast" temporal filter types (Figure 3C). We also note that these experiments used *unconstrained* spatial kernels in place of (8), yet still converged on spatial RFs with typical center-surround structure as in [10, 15, 14]. Thus, these preliminary experiments suggest that the optimal encoding strategy—in particular, the number of distinct cell types found—depends critically on the statistics of the video distribution to be encoded.

## 4.1 Mosaics fill in order of temporal frequency

As the number of RGCs available to the model increased, we observed the formation of new cell types with new spectral properties (Figure 4). We characterized the learned filters for each RGC in terms of their spectral centroid, defined as the center of mass of the Fourier (spatial) or Discrete Cosine (temporal) transform. Despite the fact that each model RGC was given its own spatial and temporal filter parameters (8, 6), the learned filter shapes strongly clustered, forming mosaics with nearly uniform response properties (Figure 4A–C). Critically, the emergence of new cell types shifted the spectral responses of previously established ones, with new cell types compressing the spectral windows of one another as they further specialized. Moreover, mosaic density increased with increasing RGC number, shifting the centroids of early mosaics toward increasingly higher spatial frequencies. This is also apparent in the forms of the typical learned filters and their power spectra: new filters selected for increasingly high-frequency content in the temporal domain (Figure 4D).

We likewise analyzed the coverage factors of both individual mosaics and the entire collection, defined as the proportion of visual space covered by the learned RFs. More specifically, we defined the spatial radius of an RF as the distance from its center at which intensity dropped to 20% of its peak and used this area to compute a coverage factor, the ratio of total RF area to total visual space ($\pi/4$ of the square's area due to circular masking). Since coverage factors depend not simply on RGC number but on RF density, they provide an alternative measure of the effective number of distinct cell types learned by the model. As Figure 4E shows, coverage increases nearly linearly with RGC number, while coverage for newly formed mosaics increases linearly before leveling off. In other words, new cell types initially increase coverage of visual space by adding new RFs, but marginal gains in coverage diminish as density increases. In all cases, the model dynamically adjusts the number of learned cell types and the proportion of RGCs assigned to them as channel capacity increases.

## 4.2 Phase changes in mosaic arrangement

In addition to retinal organization at the level of mosaics, a pair of recent papers reported both experimental [15] and theoretical [14] evidence for an additional degree of freedom in optimizing information encoding: the relative arrangement of ON and OFF mosaics. Jun et al. studied this for the case of natural images in [14], demonstrating that the optimal configuration of ON and OFF mosaics is alignment (RFs co-located) at low output noise levels and anti-alignment (OFF RFs between ON RFs and vice-versa) under higher levels of retinal output noise. Moreover, this transition is abrupt, constituting a phase change in optimal mosaic arrangement.

We thus asked whether learned mosaics exhibited a similar phase transition for natural video encoding. To do so, following [14], we repeatedly optimized a small model ($J = 14$, 7 ON, 7 OFF) for multiple learned filter types while systematically varying levels of input and output noise. In each case, one ON-OFF pair was fixed at the center of the space, while the locations of the others were allowed to vary. We used RF size $D = 8^2$ pixels for *Slow* and $D = 12^2$ for *FastA* and *FastB* cell types to allow the size of spatial kernels to be similar to those of the previous experiments, and we imposed the additional constraint that the shape parameters $a_j$, $b_j$, and $c_j$ in (8) be shared across RGCs.

Under these conditions, the six free pairs of RFs converged to either aligned (overlapping) or anti-aligned (alternating) positions along the edges of the circular visual space, allowing for a straightforward examination of the effect of input and output noises on mosaic arrangement. Figure 5A-C shows that the phase transition boundaries closely follow the pattern observed in [14]: increasing output noise shifts the optimal configuration from alignment to anti-alignment. Moreover, for each of the tested filters, increasing input noise *discourages* this transition. This effect also follows from the analysis presented in [14], since higher input noise increases coactivation of nearby pairs of RFs, requiring larger thresholds to render ON-OFF pairs approximately indpendent (Appendix B).

# 5 Discussion

**Related work:** As reviewed in the introduction, this study builds on a long line of work using efficient coding principles to understand retinal processing. In addition, it is related to work examining encoding of natural videos [25, 22, 16] and prediction in space-time. The most closely related work to this one is that of [13], which also considered efficient coding of natural videos and considered the tradeoffs involved in multiple cell types. Our treatment here differs from that work in several

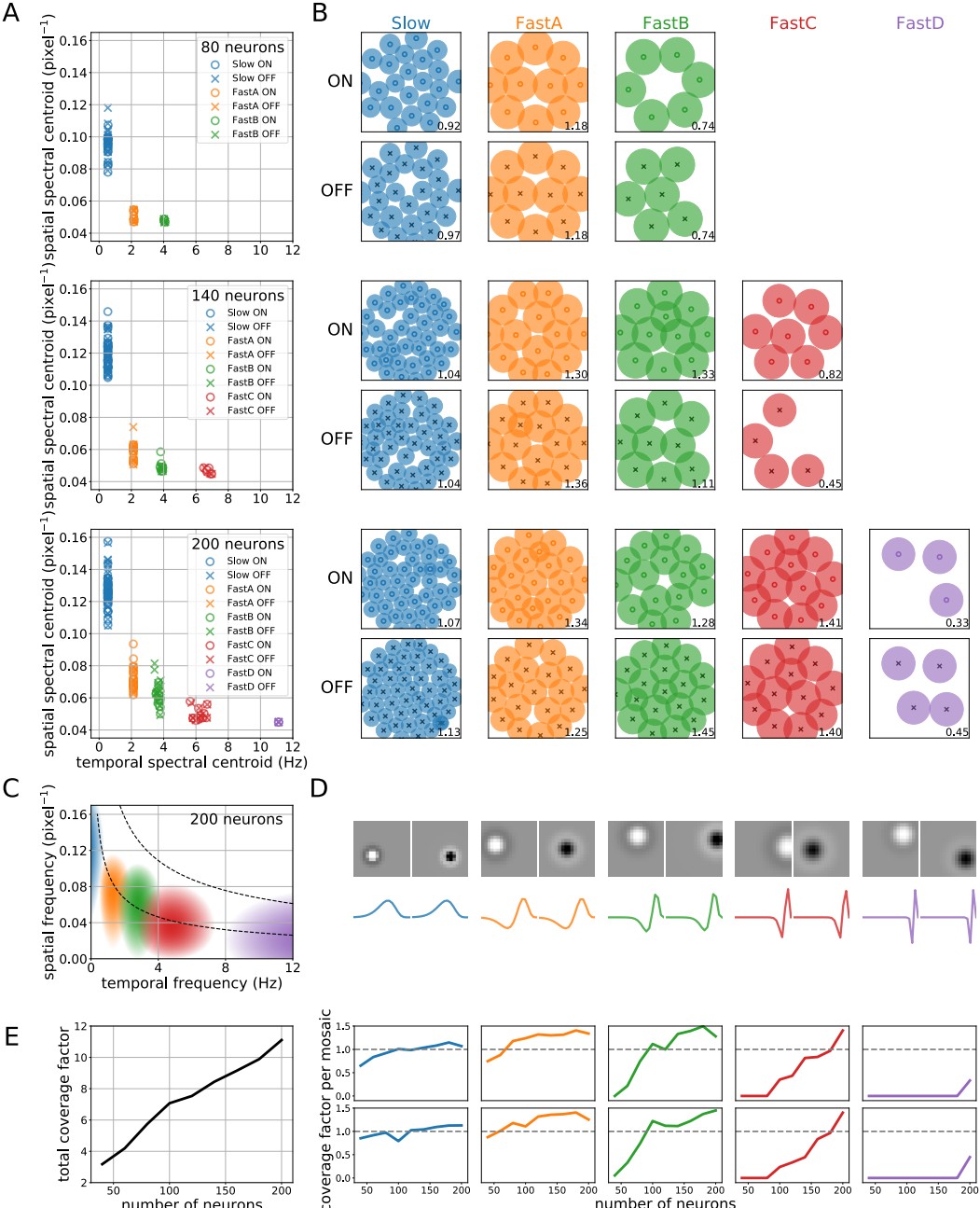

Figure 4: Emergence of new RF types with increasing RGC number. **(A)** Distribution of spatial and temporal spectral centroids for $J = 80, 140, 200$ RGCs. ON and OFF RFs form distinct clusters corresponding to different learned filter types. **(B)** ON and OFF mosaics corresponding to each cell type. The number in the lower right of each plot is the coverage factor for the mosaic. **(C)** Power spectral density of a typical kernel in each mosaic for $J = 200$. As predicted, learned kernels filter over roughly nonoverlapping regions of spatiotemporal frequency. Contour lines represent isopower lines of the signal correlation $C_x(k, \omega)$. **(D)** Learned shapes of example spatial ON and OFF filters (top) and corresponding temporal filters (bottom) from each RF type for the $J = 200$ case. **(E)** Total (left) and per-mosaic (right) coverage factors as the number of RGCs $J$ increases from 40 to 200. New mosaics increase coverage linearly with the number of RFs, while nearly full mosaics see diminishing returns in coverage from density increases. See Supplementary Figures 8-9 for similar plots for all RGC numbers.

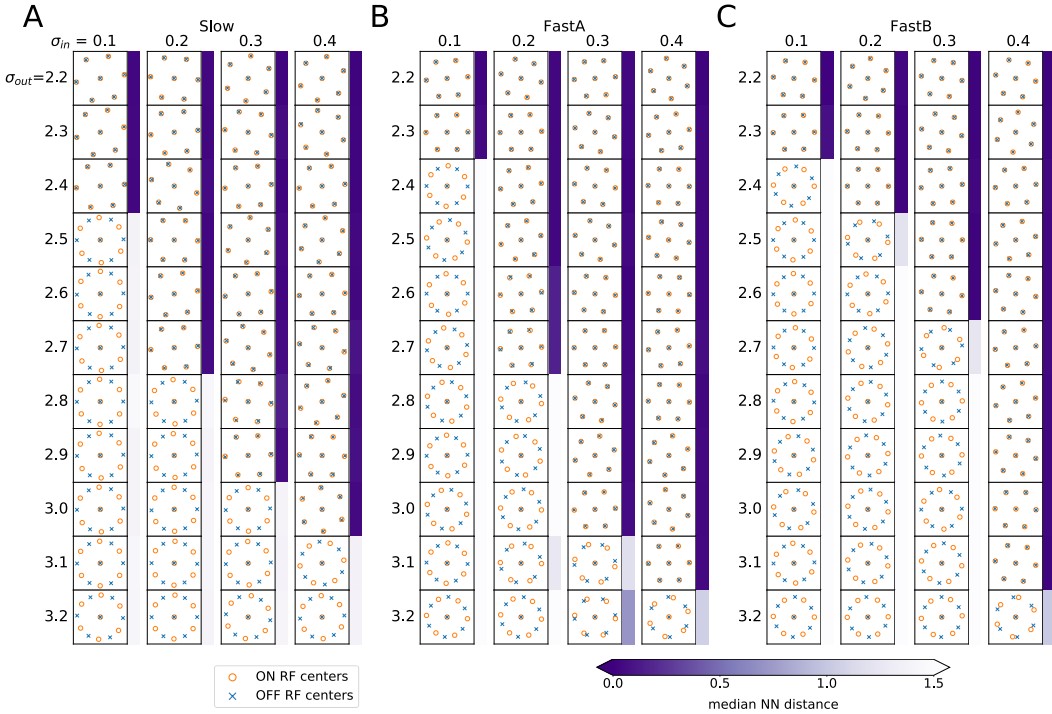

Figure 5: Learned mosaics exhibit a phase transition as a function of input and output noise. **(A-C)** Spatial kernel centers for *Slow* **(A)**, *FastA* **(B)**, and *FastB* **(C)** as a function of $\sigma_{\text{in}}$ and $\sigma_{\text{out}}$. In all three cases, the optimal configuration changes from aligned to anti-aligned when output noise increases or input noise decreases. Blue bars denote alignment as measured by median distance to the nearest RF center of the opposite type.

key ways: First, while [13] was concerned with demonstrating that multiple cell types could prove beneficial for encoding (in a framework focused on reconstruction error), that study predetermined the number of cell types and mosaic structure, only optimizing their relative spacing. By contrast, this work is focused on how the number of cell types is dynamically determined, and how the resulting mosaics arrange themselves, as a function of the number of units available for encoding (i.e., the channel capacity). Specifically, we follow previous efficient coding models [8, 9, 10, 11] in maximizing mutual information and do not assume an a priori mosaic arrangement, a particular cell spacing, or a particular number of cell types— *all of these emerge via optimization in our formulation*. Second, while the computational model of [13] optimized strides for a pair of rectangular arrays of RGCs, we individually optimize RF locations and shapes, allowing us to study changes in optimal RF size and density as new, partial mosaics begin to form. Third, while [13] used zero-padding of natural videos to bias learned temporal filters toward those of observed RGCs, we link the form of temporal RFs to biophysical limits on the filtering properties of bipolar cells, producing temporal filters with the delay properties observed in real data. Finally, while [13] only considered a single noise source in their model, we consider noise in both photoreceptor responses (input noise) and RGC responses (output noise), allowing us to investigate transitions in the optimal relative arrangement of mosaics [14, 15].

We have shown that efficient coding of natural videos produces multiple cell types with complementary RF properties. In addition, we have shown for the first time that the number and characteristics of these cell types depend crucially on the channel capacity: the number of available RGCs. As new simulated RGCs become available, they are initially concentrated into mosaics with more densely packed RFs, improving the spatial frequency bandwidth over which information is encoded. However, as this strategy produces diminishing returns, new cell types encoding higher-frequency temporal features emerge in the optimization process. These new cell types capture information over distinct spatiotemporal frequency bands, and their formation leads to upward shifts in the spatial frequency responses of previously formed cell types. Moreover, pairs of ON and OFF mosaics continue to

exhibit the phase transition between alignment and anti-alignment revealed in a purely spatial optimization of efficient coding [14], suggesting that mosaic coordination is a general strategy for increasing coding efficiency. Furthermore, despite the assumptions of this model—linear filtering, separable filters, firing rates instead of spikes—our results are consistent with observed retinal data. For example, RGCs with small spatial RFs exhibit more prolonged temporal integration: they are also more low-pass in their temporal frequency tuning. Second, there is greater variability in the size and shape of spatial RFs at a given retinal location, but temporal RFs exhibit remarkably little variability in our simulations and in data [19]. Thus, these results further testify to the power of efficient coding principles in providing a conceptual framework for understanding the nervous system.

## Acknowledgments and Disclosure of Funding

This work was supported by NIH/National Eye Institute Grant R01 EY031396.

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
