# A  Analysis details for the linear model

## A.1  Derivation of the objective in the infinite retina limit

Here, we provide details for the derivation of (9). To begin, we assume that spatial RFs centered at locations $\mathbf{z}_i$ are circularly symmetric: $w(\mathbf{z}_i, \mathbf{z}) = w(|\mathbf{z}_i - \mathbf{z}|)$. Moreover, we assume that the RF centers form a Bravais lattice $R$ with basis vectors $(\mathbf{a}_1, \mathbf{a}_2)$ in $d = 2$ such that $\mathbf{z}_i = n_{1i}\mathbf{a}_1 + n_{2i}\mathbf{a}_2$ for some integers $(n_{1i}, n_{2i})$ and the mosaic is translationally invariant under these shifts. Likewise, there exist basis vectors $(\mathbf{b}_1, \mathbf{b}_2)$ for the dual lattice $G$ such that $\mathbf{a}_i \cdot \mathbf{b}_j = 2\pi\delta_{ij}$, with the interpretation that while the RF mosaic is invariant under shifts by integer multiples of $\mathbf{a}_i$, its representation in Fourier space is invariant under shifts by the $\mathbf{b}_i$. Note that this is distinct from the assumption that the encoded images themselves are invariant under such translations. Here, for simplicity, we will transition to an infinite retina, though we will relax this assumption later.

Now, consider the determinants in (2). In the linear, discrete, purely spatial case, the numerator contains a determinant over $J \times J$ matrices with elements
$$F_{ij} = \gamma_i\gamma_j \mathbf{w}_i^\top (\mathbf{C}_x + \sigma_{\text{in}}^2 \mathbb{1})\mathbf{w}_j,$$
which can be written as
$$F_{ij} = \gamma_i\gamma_j \int d\mathbf{z}d\mathbf{z}'\, w(\mathbf{z}_i, \mathbf{z})(C_x(\mathbf{z} - \mathbf{z}') + \sigma_{\text{in}}^2\delta(\mathbf{z} - \mathbf{z}'))w(\mathbf{z}_j, \mathbf{z}')$$
$$= \int \frac{d^2\mathbf{k}}{(2\pi)^2}e^{i\mathbf{k}\cdot(\mathbf{z}_i - \mathbf{z}_j)}|v(k)|^2(C_x(k) + \sigma_{\text{in}}^2), \tag{13}$$
in the continuum limit, where we have made use of the translational and rotational symmetry of both $w$ and $C_x$ to write their Fourier transforms in terms of $k = \|\mathbf{k}\|$, and we have again defined $v \equiv \gamma w$. Of course, in the continuum limit, $J \to \infty$, and we have for a symmetric, positive-definite matrix $\mathbf{A}$
$$\log\det\mathbf{A} = \sum_i \log\lambda_i \to \int dx\,\log\lambda(x)$$
with $\lambda(x)$ the continuous eigenspectrum of $\mathbf{A}$.

Fortunately, the continuum expression (13) can be diagonalized in the Fourier basis. Let $\psi_j(\mathbf{k}') = e^{i\mathbf{k}'\cdot\mathbf{z}_j}$. Then
$$\sum_j F_{ij}\psi_j(\mathbf{k}') = \sum_j \int \frac{d^2\mathbf{k}}{(2\pi)^2}e^{i\mathbf{k}\cdot(\mathbf{z}_i - \mathbf{z}_j)}|v(k)|^2(C_x(k) + \sigma_{\text{in}}^2)e^{i\mathbf{k}'\cdot\mathbf{z}_j}$$
$$= \int \frac{d^2\mathbf{k}}{(2\pi)^2}e^{i\mathbf{k}\cdot\mathbf{z}_i}|v(k)|^2(C_x(k) + \sigma_{\text{in}}^2)\sum_j e^{i\mathbf{z}_j\cdot(\mathbf{k}'-\mathbf{k})}$$
$$= \int \frac{d^2\mathbf{k}}{(2\pi)^2}e^{i\mathbf{k}\cdot\mathbf{z}_i}|v(k)|^2(C_x(k) + \sigma_{\text{in}}^2)\cdot\text{vol}(G_0)\sum_{\mathbf{g}\in G}\delta(\mathbf{k} - \mathbf{k}' - \mathbf{g})$$
$$= \frac{\text{vol}(G_0)}{(2\pi)^2}\sum_{g\in G}|v(\mathbf{k}' + \mathbf{g})|^2(C_x(\mathbf{k}' + \mathbf{g} + \sigma_{\text{in}}^2)e^{i\mathbf{z}_i\cdot(\mathbf{k}'+\mathbf{g})}$$
$$= \left(\frac{\text{vol}(G_0)}{(2\pi)^2}\sum_{\mathbf{g}\in G}|v(\mathbf{k}' + \mathbf{g})|^2(C_x(\mathbf{k}' + \mathbf{g}) + \sigma_{\text{in}}^2)\right)e^{i\mathbf{z}_i\cdot\mathbf{k}'}$$
$$= \lambda(\mathbf{k}')\psi_i(\mathbf{k}'), \tag{14}$$
where again $G = \{p\mathbf{b}_1 + q\mathbf{b}_2 \mid p, q \in \mathbb{Z}\}$ is the dual lattice with unit cell volume $\text{vol}(G_0) = \sqrt{\det\begin{pmatrix}\|\mathbf{b}_1\|^2 & \mathbf{b}_1\cdot\mathbf{b}_2 \\ \mathbf{b}_1\cdot\mathbf{b}_2 & \|\mathbf{b}_2\|^2\end{pmatrix}}$, and $\mathbf{z}_i \cdot \mathbf{g}$ is an integer multiple of $2\pi$ by definition of the lattices $R$ and $G$. Likewise, we note that, due to aliasing from the mosaic spacing, $\psi(\mathbf{k} + \mathbf{g}) = \psi(\mathbf{k})$, so the only unique eigenvalues are those with $\mathbf{k} \in G_0$, the unit cell of the dual lattice. Thus, by a similar calculation for the denominator, the terms in (2) become
$$\int_{G_0} \frac{d^2\mathbf{k}}{(2\pi)^2}\log\frac{\frac{\text{vol}(G_0)}{(2\pi)^2}\sum_{\mathbf{g}\in G}|v(\mathbf{k} + \mathbf{g})|^2(C_x(\mathbf{k} + \mathbf{g}) + \sigma_{\text{in}}^2) + \sigma_{\text{out}}^2}{\frac{\text{vol}(G_0)}{(2\pi)^2}\sum_{\mathbf{g}\in G}|v(\mathbf{k} + \mathbf{g})|^2\sigma_{\text{in}}^2 + \sigma_{\text{out}}^2}$$

in the continuum limit.

As for the constraint term, the restriction (3) fails to generalize to the linear case, where $\mathbb{E}r_j = 0$. Instead, we note that for (1), we have

$$\mathbb{E}r_j = \gamma_j \int_{-\infty}^{\infty} dy\, \eta(y-\theta)p(y) \leq \gamma_j \sqrt{\int_{-\infty}^{\infty} dy\, \eta(y-\theta)^2 p(y)} = \gamma_j \sqrt{\mathbb{E}[\eta(y-\theta_j)^2]}$$

by Hölder's inequality. That is, we can enforce a looser restriction on firing rates by bounding the power used by the filter. Of course, in the linear case considered above, $\theta = 0$, $\eta(y) = y$, and $\mathbb{E}[y] = 0$, so that the inequality becomes

$$\mathbb{E}r_j \leq \gamma \sqrt{\mathrm{var}[\mathbf{w}^\top(\mathbf{x} + \mathbf{n}_{\mathrm{in}})]}.$$

In practice, we bound the square of this expression, which yields the continuous objective (9).

## A.2 Derivation of the optimal linear filter

As noted in Section 3.1 the optimal solution for (9) takes the form (10). However, this expression differs in two key aspects from the form originally presented in [8]: First, (10) involves a rectification operation $[\cdot]_+$ on the coefficients of the filter. Second, the integral over $\mathbf{k}$ is over the unit cell of the dual lattice $G_0$, not over the interval $[-\pi, \pi]$ (in dimension $d = 1$). Here, we provide details pertaining to both of these points.

Our starting point is (9). Here, to simplify notation, we work in dual lattice units such that $\mathrm{vol}(G_0) = (2\pi)^2$, since we can restore this later by the transformation $\sigma_{\mathrm{in}}^2 \to \sigma_{\mathrm{in}}^2 \mathrm{vol}(G_0)/(2\pi)^2$, $C_x \to C_x \mathrm{vol}(G_0)/(2\pi)^2$. Moreover, as in [8], we recognize that the optimization objective is a function only of the power spectrum $|v(\mathbf{k} + \mathbf{g})|^2$. Varying with respect to this quantity, however, requires that we enforce a positivity constraint, which implies a modified objective

$$\max_v \int_{G_0} \frac{d^2\mathbf{k}}{(2\pi)^2} \left[ \log \frac{\sum_{\mathbf{g} \in G} |v(\mathbf{k} + \mathbf{g})|^2 (C_x(\mathbf{k} + \mathbf{g}) + \sigma_{\mathrm{in}}^2) + \sigma_{\mathrm{out}}^2}{\sum_{\mathbf{g} \in G} |v(\mathbf{k} + \mathbf{g})|^2 \sigma_{\mathrm{in}}^2 + \sigma_{\mathrm{out}}^2} \right.$$
$$\left. -\nu \sum_{\mathbf{g} \in G} |v(\mathbf{k} + \mathbf{g})|^2 (C_x(\mathbf{k} + \mathbf{g}) + \sigma_{\mathrm{in}}^2) + \sum_{\mathbf{g} \in G} \alpha(\mathbf{k} + \mathbf{g})|v(\mathbf{k} + \mathbf{g})|^2 \right], \quad (15)$$

where $\alpha(\mathbf{k} + \mathbf{g})$ is a Lagrange multiplier (one per frequency) enforcing the positive-definiteness of the filter power. Taking derivatives with respect to $|v(\mathbf{k} + \mathbf{g})|^2$ and rearranging then gives:

$$(\sum_{\mathbf{g}'} |v(\mathbf{k} + \mathbf{g}')|^2 (C_x(\mathbf{k} + \mathbf{g}') + \sigma_{\mathrm{in}}^2) + \sigma_{\mathrm{out}}^2)(\sum_{\mathbf{g}''} |v(\mathbf{k} + \mathbf{g}'')|^2 \sigma_{\mathrm{in}}^2 + \sigma_{\mathrm{out}}^2)$$
$$= \frac{\sigma_{\mathrm{out}}^2 C_x(\mathbf{k} + \mathbf{g}) + \sigma_{\mathrm{in}}^2 \left[ C_x(\mathbf{k} + \mathbf{g}) \sum_{\mathbf{g}'} |v(\mathbf{k} + \mathbf{g}')|^2 - \sum_{\mathbf{g}'} C_x(\mathbf{k} + \mathbf{g}')|v(\mathbf{k} + \mathbf{g}')|^2 \right]}{\nu(C_x(\mathbf{k} + \mathbf{g}) + \sigma_{\mathrm{in}}^2) - \alpha(\mathbf{k} + \mathbf{g})} \quad (16)$$

supplemented by the complementary slackness conditions $\alpha(\mathbf{k} + \mathbf{g})|v(\mathbf{k} + \mathbf{g})|^2 = 0$.

Two things are important to note about this equation: First, if the sums over $\mathbf{g}$ are reduced to single terms (i.e., there is no power in either the correlation or filter spectra outside the unit cell of the reciprocal lattice), the term in brackets in the numerator of the right-hand side vanishes. If, in addition, $\alpha = 0$, we are back with the same expression as found in [8]. Second, the left-hand side of this equation is *the same for all* $\mathbf{g} \in G$, which implies that the right-hand side must be as well. Thus,

$$\sigma_{\mathrm{out}}^2 C_x(\mathbf{k} + \mathbf{g}) + \sigma_{\mathrm{in}}^2 \left[ C_x(\mathbf{k} + \mathbf{g}) \sum_{\mathbf{g}'} |v(\mathbf{k} + \mathbf{g}')|^2 - \sum_{\mathbf{g}'} C_x(\mathbf{k} + \mathbf{g}')|v(\mathbf{k} + \mathbf{g}')|^2 \right]$$
$$= \beta(\mathbf{k}) \left( \nu(C_x(\mathbf{k} + \mathbf{g}) + \sigma_{\mathrm{in}}^2) - \alpha(\mathbf{k} + \mathbf{g}) \right) \quad (17)$$

with $\beta(\mathbf{k}) > 0$. Recall here that, with respect to the optimization, $C_x$, $\sigma_{\mathrm{in}}^2$, and $\sigma_{\mathrm{out}}^2$ are constants and $\beta(\mathbf{k})$ is a function of $|v(\mathbf{k} + \mathbf{g})|^2$ and these constants. Only $|v(\mathbf{k} + \mathbf{g})|^2$ and $\alpha(\mathbf{k} + \mathbf{g})$ are variables

(and these are tied by complementary slackness). This suggests that the term in brackets cannot vanish for general $C_x$ unless only a single $v(\mathbf{k} + \mathbf{g}')$ in the sum is nonzero.

We check this solution in three cases:

1. Assume $v(\mathbf{k} + \mathbf{g}) \neq 0$ and $v(\mathbf{k} + \mathbf{g}') = 0$ for $\mathbf{g}' \neq \mathbf{g}$. Then the term in brackets vanishes, complementary slackness dictates $\alpha(\mathbf{k} + \mathbf{g}) = 0$, and the situation reduces to (19) below.

2. Assume $v(\mathbf{k} + \mathbf{g}) = 0$ but $v(\mathbf{k} + \mathbf{g}') \neq 0$ for a single $\mathbf{g}' \neq \mathbf{g}$. Then complementary slackness allows $\alpha(\mathbf{k} + \mathbf{g}) \neq 0$ and we have

$$\sigma_{\text{out}}^2 C_x(\mathbf{k} + \mathbf{g}) + \sigma_{\text{in}}^2 |v(\mathbf{k} + \mathbf{g}')|^2 (C_x(\mathbf{k} + \mathbf{g}) - C_x(\mathbf{k} + \mathbf{g}'))$$
$$= \beta(\mathbf{k})(\nu(C_x(\mathbf{k} + \mathbf{g}) + \sigma_{\text{in}}^2) - \alpha(\mathbf{k} + \mathbf{g})).$$

Of course, if we evaluate the right-hand side of (16) at $\mathbf{g} = \mathbf{g}'$, we get (assuming $\nu > 0$)

$$\beta(\mathbf{k}) = \frac{\sigma_{\text{out}}^2}{\nu} \frac{C_x(\mathbf{k} + \mathbf{g}')}{C_x(\mathbf{k} + \mathbf{g}') + \sigma_{\text{in}}^2},$$

and we can rewrite the optimality condition as

$$\sigma_{\text{out}}^2 C_x(\mathbf{k} + \mathbf{g}) + \sigma_{\text{in}}^2 |v(\mathbf{k} + \mathbf{g}')|^2 (C_x(\mathbf{k} + \mathbf{g}) - C_x(\mathbf{k} + \mathbf{g}'))$$
$$= \frac{\sigma_{\text{out}}^2}{\nu} \frac{C_x(\mathbf{k} + \mathbf{g}')}{C_x(\mathbf{k} + \mathbf{g}') + \sigma_{\text{in}}^2} (\nu(C_x(\mathbf{k} + \mathbf{g}) + \sigma_{\text{in}}^2) - \alpha(\mathbf{k} + \mathbf{g})).$$

If we divide both sides by $\sigma_{\text{out}}^2 C_x(\mathbf{k} + \mathbf{g}')$, we see that a solution with $\alpha(\mathbf{k} + \mathbf{g}) \geq 0$ always exists provided

$$\frac{C_x(\mathbf{k} + \mathbf{g})}{C_x(\mathbf{k} + \mathbf{g}')} + \frac{\sigma_{\text{in}}^2}{\sigma_{\text{out}}^2} |v(\mathbf{k} + \mathbf{g}')|^2 \left( \frac{C_x(\mathbf{k} + \mathbf{g})}{C_x(\mathbf{k} + \mathbf{g}')} - 1 \right) \leq \frac{C_x(\mathbf{k} + \mathbf{g}) + \sigma_{\text{in}}^2}{C_x(\mathbf{k} + \mathbf{g}') + \sigma_{\text{in}}^2}.$$

Now when $\frac{C_x(\mathbf{k}+\mathbf{g})}{C_x(\mathbf{k}+\mathbf{g}')} > 1$, the middle term in the inequality is positive but the first term on the left is larger than the right-hand side, so the inequality can never hold. By contrast, when $\frac{C_x(\mathbf{k}+\mathbf{g})}{C_x(\mathbf{k}+\mathbf{g}')} < 1$, the middle term is negative and the first term on the left is already smaller than the right-hand side, and the inequality always holds. Thus, the condition we need is that

$$C_x(\mathbf{k} + \mathbf{g}') \geq C_x(\mathbf{k} + \mathbf{g})$$

for all $\mathbf{g} \in G$. That is, the filter should respond only at the (aliased) frequency with highest power.

3. Assume $v(\mathbf{k} + \mathbf{g}) = 0$ for all $g \in G$. This happens, for instance, when the RF filter has a maximum frequency response at $k_{\max}$, which gives $v(\mathbf{k} + \mathbf{g}) = 0$ for all $\|\mathbf{k}\| > k_{\max}$. Then, from (16) first-order stationarity requires

$$\sigma_{\text{out}}^2 C_x(\mathbf{k} + \mathbf{g}) = \sigma_{\text{out}}^4 \left( \nu(C_x(\mathbf{k} + \mathbf{g}) + \sigma_{\text{in}}^2) - \alpha(\mathbf{k} + \mathbf{g}) \right)$$

or

$$\alpha(\mathbf{k} + \mathbf{g}) = (\nu - \sigma_{\text{out}}^{-2}) C_x(\mathbf{k} + \mathbf{g}) + \nu \sigma_{\text{in}}^2. \tag{18}$$

Together, these calculations suggest the following pattern in the filter response $v(\mathbf{k} + \mathbf{g})$: Assuming that $C_x(\mathbf{k})$ is a monotonically decreasing function of $k = \|\mathbf{k}\|$, the filter can only respond for at most *one* aliased value of the frequency $\mathbf{k}$. The optimal frequency at which to respond is, from point 2 above, the one with the highest power, and given our assumption about $C_x(\mathbf{k})$, this is the one with the *lowest frequency*. As a result, $v(\mathbf{k})$ only has support within the unit cell $G_0$, the sums over $\mathbf{g} \in G$ vanish, and at these frequencies, the solution is given by the term inside the brackets in (10). Moreover, at values of $\mathbf{k}$ for which the term inside the brackets in (10) would be 0 or negative, the correct solution is given by $v(\mathbf{k} + \mathbf{g}) = 0$ with $\alpha(\mathbf{k} + \mathbf{g})$ given by (18). Thus, returning to (16), we have

$$(|v(k)|^2 (C_x(k) + \sigma_{\text{in}}^2) + \sigma_{\text{out}}^2)(|v(k)|^2 \sigma_{\text{in}}^2 + \sigma_{\text{out}}^2) = \frac{\sigma_{\text{out}}^2 C_x(k)}{\nu(C_x(k) + \sigma_{\text{in}}^2) - \alpha(k)}, \tag{19}$$

and collecting terms gives

$$|v(k)|^4 + |v(k)|^2 \frac{\sigma_{\text{out}}^2}{\sigma_{\text{in}}^2}\left(1 + \frac{\sigma_{\text{in}}^2}{C_x(k) + \sigma_{\text{in}}^2}\right) + \frac{\sigma_{\text{out}}^2}{\sigma_{\text{in}}^2(C_x(k) + \sigma_{\text{in}}^2)}\left[\sigma_{\text{out}}^2 - \frac{C_x(k)}{\nu'(C_x(k) + \sigma_{\text{in}}^2)}\right] = 0$$

with $\nu' \equiv \nu - \frac{\alpha(k)}{C_x(k) + \sigma_{\text{in}}^2}$. This has the solution

$$|v(k)|^2 = -\frac{1}{2}\frac{\sigma_{\text{out}}^2}{\sigma_{\text{in}}^2}\left(1 + \frac{\sigma_{\text{in}}^2}{C_x(k) + \sigma_{\text{in}}^2}\right) + \frac{1}{2}\frac{\sigma_{\text{out}}^2}{\sigma_{\text{in}}^2}\frac{C_x(k)}{C_x(k) + \sigma_{\text{in}}^2}\sqrt{1 + \frac{\sigma_{\text{in}}^2}{\sigma_{\text{out}}^2}\frac{4}{\nu' C_x(k)}}$$

$$= \frac{\sigma_{\text{out}}^2}{\sigma_{\text{in}}^2}\left[\frac{1}{2}\frac{C_x(k)}{C_x(k) + \sigma_{\text{in}}^2}\left(\sqrt{1 + \frac{\sigma_{\text{in}}^2}{\sigma_{\text{out}}^2}\frac{4}{\nu' C_x(k)}} + 1\right) - 1\right]$$

When the term in brackets is positive, complementary slackness requires $\alpha(k) = 0$ and thus $\nu' = \nu$, while when the bracketed term is less than or equal to zero, $\alpha(k)$ need not be zero, but complementary slackness requires $|v(k)|^2 = 0$, which holds when $\alpha(k)$ satisfies (18). Thus, the full solution is given by (10).

### A.3   Extension to the spatiotemporal case

Consider the same setup as in the previous section but with a spacetime filter $\gamma w(\mathbf{z}, t) = v(\mathbf{z}, t)$. Now, the relevant correlations inside the determinant, analogous to (13), are between firing rates at different filters at different times:

$$F_{ijtt'} = \int d\mathbf{z}d\mathbf{z}'d\tau d\tau' \gamma^2 w(\mathbf{z}_i - \mathbf{z}, t - \tau)C_x(\mathbf{z} - \mathbf{z}', \tau - \tau')w(\mathbf{z}_j - \mathbf{z}', t' - \tau')$$

$$= \int \frac{d^2\mathbf{k}}{(2\pi)^2}\frac{d\omega}{2\pi}e^{i\mathbf{k}\cdot(\mathbf{z}_i - \mathbf{z}_j) + i\omega(t - t')}|v(\mathbf{k}, \omega)|^2 C_x(\mathbf{k}, \omega),$$

and once again we can diagonalize this by taking eigenvectors

$$\psi_{jt}(\mathbf{k}', \omega') = e^{i\mathbf{z}_j \cdot \mathbf{k}' + i\omega' t}$$

for which

$$\sum_j \int dt' F_{ijtt'}\psi_{jt'}(\mathbf{k}', \omega') = \lambda(\mathbf{k}', \omega')\psi_{it}(\mathbf{k}', \omega').$$

Now, just as in the spatial domain, this leads to an objective in the Fourier domain

$$\max_v \int_{G_0}\frac{d^2\mathbf{k}}{(2\pi)^2}\int\frac{d\omega}{2\pi}\left[\log\frac{\sum_{\mathbf{g}\in G}|v(\mathbf{k} + \mathbf{g}, \omega)|^2(C_x(\mathbf{k} + \mathbf{g}, \omega) + \sigma_{\text{in}}^2) + \sigma_{\text{out}}^2}{\sum_{\mathbf{g}\in G}|v(\mathbf{k} + \mathbf{g}, \omega)|^2\sigma_{\text{in}}^2 + \sigma_{\text{out}}^2}\right.$$

$$\left. -\nu\sum_{\mathbf{g}\in G}|v(\mathbf{k} + \mathbf{g}, \omega)|^2(C_x(\mathbf{k} + \mathbf{g}, \omega) + \sigma_{\text{in}}^2) + \sum_{\mathbf{g}\in G}\alpha(\mathbf{k} + \mathbf{g}, \omega)|v(\mathbf{k} + \mathbf{g}, \omega)|^2\right]. \quad (20)$$

Clearly, by the same arguments as above, the solution to this is once again (10) with $v(k) \to v(k, \omega)$, subject to a normalization condition over both spatial and temporal frequencies.

However, the solution given in (10) only determines the power spectrum of the optimal filter, not its phase. That is, if $v(\mathbf{k}, \omega) = |v(\mathbf{k}, \omega)|e^{i\phi(\mathbf{k}, \omega)}$, $\phi(\mathbf{k}, \omega)$ is undetermined. However, for the minimum-phase system, which is both causal and imposes the minimum temporal delay between the incoming signal and its filtered response [20], there is a relation between the filter's amplitude $|v(\mathbf{k}, \omega)|$ (treating $\mathbf{k}$ as constant) and its phase $\phi(\mathbf{k}, \omega)$:

$$\phi(\mathbf{k}, \omega) = -\mathcal{H}[\log|v(\mathbf{k}, \omega)|], \quad (21)$$

where $\mathcal{H}[\cdot]$ is the Hilbert transform (over $\omega$) [26]. More explicitly, the minimum phase filter has

$$\log|v(\mathbf{k}, \omega)| + i\phi(\mathbf{k}, \omega) = \log|v(\mathbf{k}, \omega)| - i\mathcal{H}[\log|v(\mathbf{k}, \omega)|], \quad (22)$$

which is the complex conjugate of the analytic signal and so is analytic in the *lower* half plane (and thus causal). For a given, fixed $\mathbf{k}$, this can be used to plot the causal temporal filters as depicted in Figure 2C. Note also that the spatiotemporal filter found by generalizing (10) is not exactly separable, though it may be well approximated by the product of a spatial and temporal filter in certain regimes.

Finally, we note that the solution (19) in both its spatial and spatiotemporal forms exhibits a kink in its power spectral density stemming from the positive rectification. This in turn implies that the optimal spatial filters exhibit ringing in both the space and time domains, unlike the actual observed retinal filters. Of course, these solutions, like many ideal filters, are all but impossible to implement in real systems, necessitating tradeoffs among ripple, rolloff, and other attributes [26]. Indeed, as discussed above, allowable temporal filters are constrained by bipolar cell response properties [20]. As a result, in Figure 2B-D, we have smoothed the optimal power spectral densities with a double exponential kernel $e^{-a\lceil k \rceil}$ $(e^{-a|\omega|})$ prior to transforming back to the space (time) domain.

## A.4  Independence of multiple filters

Here, we generalize the approach given in Appendix A.1 to multiple filter types, arguing that requiring these filters not to overlap in frequency space is sufficient to optimize the objective 2 in the infinite retina limit. The only relevant difference between this case and that of Appendix A.1 is that, in addition to terms like (13), the determinants also contain cross-terms of the form

$$ F_{ijtt'} = \int \frac{d^2\mathbf{k}}{(2\pi)^2} \frac{d^2\omega}{(2\pi)} e^{i\mathbf{k}\cdot(\mathbf{z}_i - \mathbf{z}_j) + i\omega(t - t')} v^*(k,\omega) v'(k,\omega) (C_x(k) + \sigma_{\text{in}}^2). \tag{23} $$

The presence of such cross-terms, which arises from correlations between activity in the two mosaics, reduce the magnitude of the determinants and thus overall information. However, we argue that such cross-terms should vanish for the optimal solution, yielding an information objective that is equivalent to a sum of individual mosaic terms like (9). Intuitively, this is because our power constraint in (9) remains unchanged under an orthogonal transformation of the filters, while information is *subadditive* when cross-terms between filters are nonzero. Thus, we can always increase information by choosing a filter basis in which both $\mathbf{C}_x$ and $\mathbf{C}_{\text{in}}$ are block diagonal and the mosaics decouple. This logic is similar to the argument of [13], where it was reconstruction performance that was equivalent under transformations of the filters but costs were superadditive.

However, this reasoning does not fully fix the choice of independent filters, since there are multiple ways to make off-diagonal blocks in the determinants vanish. For example, in the discrete (pixel, frame) formulation of the problem, the generalized eigenvectors of $(\mathbf{C}_x, \mathbf{C}_{\text{in}})$ represent a special solution (the one that *fully* diagonalizes both matrices). Of course, these filters need not optimize the information objective. But there is an alternative solution that removes only the off-diagonal blocks in the determinants, reducing the problem to one of again optimizing individual filters with the objective (9): choosing spectrally disjoint filters with $v^*(k,\omega) v'(k,\omega) = 0$ for different mosaics defined by $v$ and $v'$. Note that this condition is merely sufficient, not necessary, to decouple mosaics, but it is independent of any assumptions on the image correlation structure $\mathbf{C}_x$ or the filters $v(k,\omega)$. More specifically, it does not depend on any assumptions of spacetime separability for either.

In [13], the authors considered a form of this same bandwidth partitioning in the spatial case $(v^*(k) v'(k) = 0)$, but as we show Appendix A.5, this approach encounters a limit beyond the first two mosaics, when the spatial passband spectra of new filters substantially overlap. Rather, the more general solution is that mosaic filters are band-limited in both space and time, arranging themselves to tile minimally overlapping regions with highest power in the $(k,\omega)$ plane, as we find in the full nonlinear model (Figure 4).

## A.5  Effects of channel capacity and new mosaic formation

Now we reconsider the analysis of Appendices A.1 and A.3 in the case that the number of RGCs $J$ in the system is varied. To do so, we take a *finite* retina of size $(L_1, L_2) = (M_1 \|\mathbf{a}_1\|, M_2 \|\mathbf{a}_2\|)$ along each basis vector of the lattice $R$ and assume a periodic extension of the signal outside this domain. This approximation allows us to continue working in the Fourier domain but sets a *lower* bound on the spatial frequencies a mosaic can hope to resolve (in addition to the *upper* bound resulting from the mosaic lattice spacing). Note that for a fixed retina size, increasing $M_i$ is equivalent to reducing $\|\mathbf{a}_i\|$, i.e., decreasing the spacing between RF centers.

More explicitly, let $\mathbf{z}_j = n_{1j}\mathbf{a}_1 + n_{2j}\mathbf{a}_2$ as before, with RGC number $J \approx \mathrm{vol}(R)/\mathrm{vol}(R_0) = M_1 M_2$. That is, the number of RGCs is the size of the retina divided by the unit cell volume of the Bravais lattice. In addition, the assumed periodicity of the signals implies that Fourier transforms involve only frequencies of the form $\mathbf{k}_m = \frac{m_1}{M_1}\mathbf{b}_1 + \frac{m_2}{M_2}\mathbf{b}_2$ for $m_i \in \mathbb{Z}$. From this, following the derivation preceding (14), we have that $\psi_j(\mathbf{k}_l) = e^{i\mathbf{z}_j \cdot \mathbf{k}_l}$ are eigenvectors of the $J \times J$ matrix $F_{ij}$:

$$
\begin{aligned}
\sum_j F_{ij}\psi_j(\mathbf{k}_l) &= \frac{1}{\mathrm{vol}(R)} \sum_j \sum_m e^{i\mathbf{k}_m \cdot (\mathbf{z}_i - \mathbf{z}_j)} |v(\mathbf{k}_m)|^2 (C_x(\mathbf{k}_m) + \sigma_{\mathrm{in}}^2) e^{i\mathbf{z}_j \cdot \mathbf{k}_l} \\
&= \frac{1}{\mathrm{vol}(R)} \sum_m e^{i\mathbf{k}_m \cdot \mathbf{z}_i} |v(\mathbf{k}_m)|^2 (C_x(\mathbf{k}_m) + \sigma_{\mathrm{in}}^2) \sum_j e^{i\mathbf{z}_j \cdot (\mathbf{k}_l - \mathbf{k}_m)} \\
&= \frac{1}{\mathrm{vol}(R)} \sum_m e^{i\mathbf{k}_m \cdot \mathbf{z}_i} |v(\mathbf{k}_m)|^2 (C_x(\mathbf{k}_m) + \sigma_{\mathrm{in}}^2) \, J \sum_{\mathbf{g} \in G} \delta_{\mathbf{k}_l, \mathbf{k}_m + \mathbf{g}} \\
&= \frac{J}{\mathrm{vol}(R)} \left( \sum_{\mathbf{g} \in G} |v(\mathbf{k}_l + \mathbf{g})|^2 (C_x(\mathbf{k}_l + \mathbf{g}) + \sigma_{\mathrm{in}}^2) \right) e^{i\mathbf{k}_l \cdot \mathbf{z}_i} \\
&= \frac{\mathrm{vol}(G_0)}{(2\pi)^2} \left( \sum_{\mathbf{g} \in G} |v(\mathbf{k}_l + \mathbf{g})|^2 (C_x(\mathbf{k}_l + \mathbf{g}) + \sigma_{\mathrm{in}}^2) \right) e^{i\mathbf{k}_l \cdot \mathbf{z}_i},
\end{aligned}
$$

since $\mathrm{vol}(R_0) \cdot \mathrm{vol}(G_0) = (2\pi)^2 \Rightarrow J/\mathrm{vol}(R) = 1/\mathrm{vol}(R_0) = \mathrm{vol}(G_0)/(2\pi)^2$. Clearly, this is the analogue of (14), and the form of the optimal spatial filter 10 once again holds in spacetime, albeit only at a finite set of frequencies $\mathbf{k}_m \in G_0$. In what follows, we will be interested in the effect of changing $J$ on the information encoded by the filters.

Turning to the images themselves, we here consider an idealized form of the power spectrum of natural videos [21],

$$
C_x(k, \omega) \simeq \frac{A}{k^\alpha \omega^2} \tag{24}
$$

where $\alpha \approx 1.3$. Of course, this is not the same as the spectrum of images presented to the retina, since the latter is low-pass filtered via the modulation transfer function of the eye [9], but here we analyze this simpler form, since the qualitative results are similar in both cases. For this spectrum, the optimal filter (10) takes the form

$$
|v(k, \omega)|^2 = \frac{\mathrm{vol}(R)}{J} \frac{\sigma_{\mathrm{out}}^2}{\sigma_{\mathrm{in}}^2} \left[ \frac{1}{2} \frac{A}{A + \sigma_{\mathrm{in}}^2 k^\alpha \omega^2} \left( \sqrt{1 + \frac{\sigma_{\mathrm{in}}^2}{\sigma_{\mathrm{out}}^2} \frac{4k^\alpha \omega^2}{\nu A} + 1} \right) - 1 \right]_+ .
$$

Now, we will define two useful frequencies in terms of our parameters

$$
k_f(\omega) \equiv \left( \frac{A}{\sigma_{\mathrm{in}}^2 \omega^2} \right)^{\frac{1}{\alpha}} \tag{25}
$$

$$
k_c(\omega) \equiv \left( \frac{A}{\sigma_{\mathrm{in}}^2 \omega^2 \nu \sigma_{\mathrm{out}}^2} \right)^{\frac{1}{\alpha}} = \frac{k_f}{(\nu \sigma_{\mathrm{out}}^2)^{1/\alpha}}. \tag{26}
$$

Often, $\nu \sigma_{\mathrm{out}}^2 \ll 1$, so that $k_c \gg k_f$. In terms of these, the filter can be rewritten

$$
|v(k, \omega)|^2 \propto \left[ \frac{1}{2} \frac{1}{1 + \tilde{k}(\omega)} \left( \sqrt{1 + \frac{4}{\nu \sigma_{\mathrm{out}}^2} \tilde{k}(\omega)} + 1 \right) - 1 \right]_+
$$

with $\tilde{k} \equiv (k/k_f(\omega))^\alpha$. That is, up to an overall rescaling, the filter depends only on the ratio $\tilde{k}$ and the dimensionless parameter $\nu \sigma_{\mathrm{out}}^2$. Naively, changes in $k_f$ simply rescale the filter in space, while $\nu$ is related to the power consumed by the filter. However, given a *fixed* power budget, these two parameters are tied, leading to a single family of spatial filters that depend on the temporal frequency $\omega$ (and vice-versa).

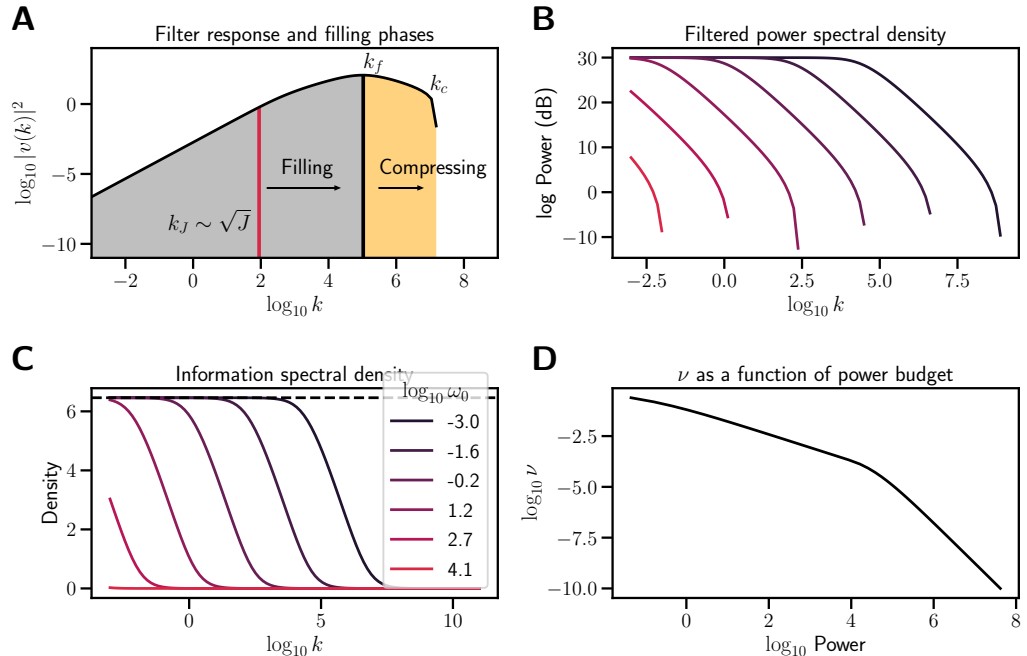

Supplementary Figure 1: Information as a function of bandwidth. (**A**) Schematic of the optimal filter. The optimal filter response comprises a log-linear region for $k < k_f$ and a negative log-linear region for $k_f < k < k_c$. As RGC number $J \sim k^2$ increases through the former, the mosaic fills, pushing the bandwidth limit (red line) upward. When the mosaic is filled, the upper bandwidth limit continues to increase as new RFs pack into the fixed retinal space and compress the mosaic. (**B**) Optimal filter power spectral density for different values of characteristic temporal response frequency $\omega_0$. Brighter colors indicate increasing frequency (see legend in **C**). In each case, the filtered spectrum is flat and proportional to $\nu^{-1}$ for $k < k_f$. For $k > k_f$, the power is noise-dominated, with a $k^{-\alpha/2}$ tail. (**C**) Information as a function of frequency for the same setup as in **B**. Note that information content falls precipitously for $k > k_f$ (the vertical axis is not logged). The dotted line indicates $-\log \nu \sigma_{\text{out}}^2$. (**D**) As the power budget increases, $\nu$ decreases, with $\nu \sim P^{-1}$ at low power and $\nu \sim P^{-2}$ as $P \to \infty$. For all plots, $A = 100$, $\sigma_{\text{in}} = 0.4$, $\sigma_{\text{out}} = 1.25$. In **A–C**, $\nu = 10^{-3}$. In **A**, $\log_{10} \omega_0 = -2.3$; in **D**, $\log_{10} \omega_0 = 1.2$.

We can gain additional insight by examining two important regimes in this distribution: If $k \gtrsim k_f$ then

$$|v(k,\omega)|^2 \approx \frac{\text{vol}(R)}{J} \frac{\sigma_{\text{out}}^2}{\sigma_{\text{in}}^2} \left[ \left( \frac{k_c(\omega)}{k} \right)^{\alpha/2} - 1 \right]_+ , \qquad C_x(k,\omega) + \sigma_{\text{in}}^2 \approx \sigma_{\text{in}}^2, \qquad (27)$$

consistent with the definition of $k_c$ as an $\omega$-dependent cutoff frequency . Conversely, when $k \lesssim k_f(\omega)$,

$$|v(k,\omega)|^2 \approx \frac{k^\alpha \omega^2}{A\nu}, \qquad (28)$$

independent of $J$, the filter is separable, and the image spectrum is white [9]:

$$|v(k,\omega)|^2 C_x(k,\omega) \approx \frac{1}{\nu}.$$

Supplementary Figure 1A shows that indeed, the log-scaled filter magnitude is characterized by a linear regime when $k \lesssim k_f$, followed by a $\frac{1}{k^{\alpha/2}}$ decay when $k \gtrsim k_f$. Thus, despite the fact that power density is highest in the first regime, the second is light-tailed and stretches exponentially longer, with the result that total power is dominated by the tail (Supplementary Figure 1B). And indeed, numerical integration shows that power initially scales as $\nu^{-1}$ for large values, followed by $P \sim \nu^{-1/2}$ as $\nu \to 0$, consistent with this claim (Supplementary Figure 1D). By contrast, as

Supplementary Figure 1C shows, information density (log terms in (20)) falls off much more rapidly with frequency, such that it is the $k \lesssim k_f$ regime that dominates, and information $\mathcal{I} \approx -\log \frac{\nu}{\sigma_{\text{out}}^2}$.

However, the above analysis is all in the continuous (infinite retina case). For the finite retina case, we note two important changes: First, as the number of RGCs increases, so does the power budgeted to the system. That is, the allowed power is assumed to be $P(J) = P_0 J$. This need not be the case — some other scaling could just as well be chosen — but it is consistent with the idea that metabolic costs are driven by factors specific to each cell. Second, the information in the system is no longer given by the the first term in the integral (20) but by a sum over discrete frequencies $\mathbf{k}_m \in \tilde{G}_0 \equiv G_0 \cap \{\frac{m_1}{M_1}\mathbf{b}_1 + \frac{m_2}{M_2}\mathbf{b}_2 \mid m_i \in \mathbb{Z}\} = \{\frac{m_1}{M_1}\mathbf{b}_1 + \frac{m_2}{M_2}\mathbf{b}_2 \mid m_i = 0 \dots M_i - 1\}$. For example, when $J = 1$, $M_1 = M_2 = 1$ and $m_1 = m_2 = 0$ is the only option — the periodicity of the signals is the periodicity of the lattice — whereas $J = 2$ allows either $M_1 = 2$ or $M_2 = 2$ but not both, and $J = 4$ has $M_1 = M_2 = 2$. In general, for large enough $J$, we expect $k_m = \|\mathbf{k}_m\| \sim \sqrt{J}$.

With these conventions, we then have modified expressions for both the information in the system and its power consumption:

$$\mathcal{I} = \sum_j \log \frac{\left[\frac{1}{2}\frac{A}{\sigma_{\text{in}}^2 k_j^\alpha \omega_0^2}\left(\sqrt{1 + \frac{\sigma_{\text{in}}^2}{\sigma_{\text{out}}^2}\frac{4k_j^\alpha \omega^2}{\nu A}} + 1\right) - \frac{A}{\sigma_{\text{in}}^2 k_j^\alpha \omega_0^2} - 1\right]_+ + 1}{\left[\frac{1}{2}\frac{A}{A + \sigma_{\text{in}}^2 k_j^\alpha \omega_0^2}\left(\sqrt{1 + \frac{\sigma_{\text{in}}^2}{\sigma_{\text{out}}^2}\frac{4k_j^\alpha \omega^2}{\nu A}} + 1\right) - 1\right]_+ + 1}$$

$$= \sum_j \log \frac{\left[\frac{1}{2\tilde{k}_j}\left(\sqrt{1 + \frac{4\tilde{k}_j}{\nu \sigma_{\text{out}}^2}} - 1\right) - 1\right]_+ + 1}{\left[\frac{1}{2(1+\tilde{k}_j)}\left(\sqrt{1 + \frac{4\tilde{k}_j}{\nu \sigma_{\text{out}}^2}} + 1\right) - 1\right]_+ + 1} = \sum_j \log \frac{P_j + \sigma_{\text{out}}^2}{\frac{\tilde{k}_j}{1+\tilde{k}_j}P_j + \sigma_{\text{out}}^2} \qquad (29)$$

$$P = P_0 J = \sigma_{\text{out}}^2 \sum_j \left[\frac{1}{2\tilde{k}_j}\left(\sqrt{1 + \frac{4\tilde{k}_j}{\nu \sigma_{\text{out}}^2}} - 1\right) - 1\right]_+ = \sum_j P_j, \qquad (30)$$

where we have chosen to focus on filters narrowly concentrated around a single frequency $\omega_0$ and again used $\tilde{k} \equiv (k/k_f(\omega_0))^\alpha$.

Several things are important to note about the scaling relationships in (29) and (30). First, as discussed above, for $\tilde{k} \lesssim 1$ ($k \lesssim k_f$),

$$P(k) + \sigma_{\text{out}}^2 \approx \nu^{-1} \approx P_0 + \sigma_{\text{out}}^2,$$

independent of $k$. That is, contributions to (30) are all roughly the same for each frequency. Similarly, the denominator in (29) is $\frac{\tilde{k}_j}{\tilde{k}_j+1}P_j + \sigma_{\text{out}}^2 \approx \sigma_{\text{out}}^2 + \tilde{k}_j P_j$ to lowest order in $\tilde{k}$, giving

$$\mathcal{I} \approx J \log\left(1 + \frac{P_0}{\sigma_{\text{out}}^2}\right) - \frac{P_0}{\sigma_{\text{out}}^2}\sum_j \tilde{k}_j \approx J\left[\log\left(1 + \frac{P_0}{\sigma_{\text{out}}^2}\right) - \frac{2P_0}{(\alpha+2)\sigma_{\text{out}}^2}\left(\frac{J}{J_f}\right)^{\frac{\alpha}{2}}\right], \qquad (31)$$

where we have used the volume equivalence $k_J^2/k_f^2 = J/J_f$ along with the integral approximation

$$\sum_j \tilde{k}_j \approx \frac{1}{V}\int_0^{k_J} dk\, k\left(\frac{k}{k_f}\right)^\alpha = \frac{2}{V(2+\alpha)}\frac{k_J^{\alpha+2}}{k_f^\alpha}$$

$$J = \frac{1}{V}\int_0^{k_J} dk\, k \quad \Rightarrow \quad V = \frac{k_J^2}{2J}.$$

We can understand this $k \lesssim k_f$ regime as a *mosaic filling phase*: For small energy budgets $P_0 \ll \sigma_{\text{out}}^2$, new RFs are added at their preferred size ($k_{\text{peak}}^{-1} \approx k_f^{-1}$) until the spacing between RFs reaches a minimum of roughly $2\pi/k_f$ (at which point $C_x(k) \approx \sigma_{\text{in}}^2$), with information increasing almost linearly as the mosaic covers new spatial locations. Whereas, for larger energy budgets $P_0 \gtrsim \sigma_{\text{out}}^2$, information gain is sublinear in $J$ with a correction term $\sim J^{\alpha/2}$.

By contrast, for $k \gtrsim k_f$, the system enters a *mosaic compression phase*. In this regime, noise dominates signal, $P_j + \sigma_{\text{out}}^2 \sim \sigma_{\text{out}}^2 (\frac{k_c(\nu)}{k_j})^{\alpha/2}$, and for the power above $k_f$, we have

$$(P_0 + \sigma_{\text{out}}^2)(J - J_f) = \sum_{J=J_f}^{J} \frac{\sigma_{\text{out}}^2}{\sqrt{\sigma_{\text{out}}^2 \nu}} \left(\frac{k_f}{k_j}\right)^{\frac{\alpha}{2}}$$

$$\approx \frac{1}{V} \int_{k_f}^{k_J} dk\, k\, \frac{\sigma_{\text{out}}}{\sqrt{\nu}} \left(\frac{k_f}{k}\right)^{\frac{\alpha}{2}}$$

$$= \frac{1}{V} \frac{\sigma_{\text{out}}}{\sqrt{\nu}} \frac{k_f^{\frac{\alpha}{2}}}{2 - \frac{\alpha}{2}} \left(k_J^{2-\frac{\alpha}{2}} - k_f^{2-\frac{\alpha}{2}}\right)$$

where $J_f$ is the number of RGCs contained in $k \lesssim k_f$ and $V$ is once again a normalizing factor required to convert the sum over discrete frequencies into the integral over $k$:

$$\frac{1}{V} \int_{k_f}^{k_J} dk\, k = (J - J_f) \quad \Rightarrow \quad V = \frac{k_J^2 - k_f^2}{2(J - J_f)}.$$

Putting these together and using $k_J^2/k_f^2 = J/J_f$ in two dimensions then gives

$$(P_0 + \sigma_{\text{out}}^2)(J - J_f) \approx (J - J_f) \frac{2}{2 - \frac{\alpha}{2}} \frac{\sigma_{\text{out}}}{\sqrt{\nu}} \frac{\left(\frac{J}{J_f}\right)^{1-\frac{\alpha}{4}} - 1}{\left(\frac{J}{J_f}\right) - 1}$$

$$\xrightarrow{J \gg J_f} (J - J_f) \frac{2}{2 - \frac{\alpha}{2}} \frac{\sigma_{\text{out}}}{\sqrt{\nu}} \left(\frac{J}{J_f}\right)^{-\frac{\alpha}{4}}, \tag{32}$$

which requires $\nu \sim (J/J_f)^{-\alpha/2}$ asymptotically for proper power scaling.

From (29), the compression phase adds then adds information

$$\mathcal{I}_{\text{compression}} = \sum_{j=J_f}^{J} \log \frac{P_j + \sigma_{\text{out}}^2}{\frac{\tilde{k}_j}{\tilde{k}_j+1} P_j + \sigma_{\text{out}}^2} \approx \sum_{j=J_f}^{J} \log \frac{1 + \tilde{k}_j}{\tilde{k}_j + \sqrt{\nu \sigma_{\text{out}}^2 \tilde{k}_j}}$$

$$\xrightarrow{k \ll k_c} \sum_{j=J_f}^{J} \log \left(1 + \frac{1}{\tilde{k}_j}\right) \approx \sum_{j=J_f}^{J} \frac{1}{\tilde{k}_j}$$

$$\approx \frac{1}{V} \int_{k_f}^{k_J} dk\, k \left(\frac{k_f}{k}\right)^{\alpha} = (J - J_f) \frac{2}{2 - \alpha} \frac{\left(\frac{J}{J_f}\right)^{1-\frac{\alpha}{2}} - 1}{\left(\frac{J}{J_f}\right) - 1}$$

$$\xrightarrow{J \gg J_f} (J - J_f) \left(\frac{J}{J_f}\right)^{-\frac{\alpha}{2}} \frac{2}{2 - \alpha}, \tag{33}$$

which is again sublinear growth in $J$, but not so strong as the decay in (31) when $P_0 \gtrsim \sigma_{\text{out}}^2$.

Finally, we consider the effects of temporal frequency $\omega$ on the relationships we have described above. In the case we have assumed, that of a filter with a narrow temporal passband centered around $\omega_0$, we note that (29) depends only on $\tilde{k}_j$, and we have

$$\tilde{k}_j = \left(\frac{k_j}{k_f}\right)^{\alpha} = \frac{k_j^{\alpha} \sigma_{\text{in}}^2 \omega_0^2}{A}.$$

Thus, if we increase $\omega_0 \to \theta \omega_0$ with $\theta > 1$, we have $\tilde{k} \to \theta^2 \tilde{k}$, which moves the numerator and denominator in (29) closer to each other and information to 0. The effect of this change in temporal frequency can be seen in Figure 2E. As a result, mosaics with the lowest temporal frequency are filled first, with new mosaics only beginning to fill once the marginal information gain for adding the first RF to a new mosaic exceeds that of adding an additional RF to an existing mosaic. In this way, the rate of each mosaic's filling decreases as new mosaics are added Figure 2F.

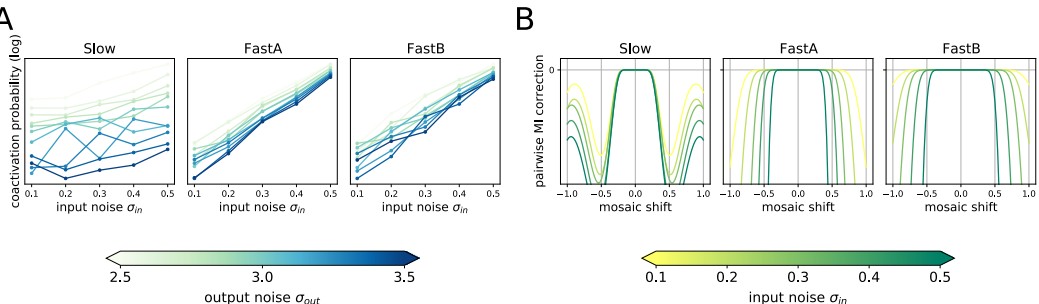

Supplementary Figure 2: **(A)** Across all output noise levels, the coactivation probability of an ON RGC at the center and an OFF RGC at the edge increases as the input noise $\sigma_{\text{in}}$ increases. **(B)** The pairwise mutual information correction term, which captures the reduction in mutual information due to pairwise coactivations of RFs (plotted for $\sigma_{\text{out}} = 3.0$) stays near zero for a shorter range of mosaic shifts at higher input noise levels, indicating that higher input noise favors aligned mosaics.

## B    Analysis of mosaic phase transitions as a function of input noise

Figure 5 in the main text shows that, for each filter type, increasing input noise levels encourage alignment. That is, the transition happens at a higher output noise level. Here, we explain this seemingly paradoxical effect using the mathematical analysis presented in [14], whose notation we follow. In that work, it was argued that the transition from aligned to anti-aligned mosaics could be understood as a process of increasing output noise leading to higher response thresholds, which in turn decorrelated nearby ON and OFF cells [12, 14]. As these pairs become more independent (as measured by $p_2$, their pairwise coactivation probability), they remain so even for small relative shifts in position, implying that anti-alignment does not increase coding redundancy. Add to this the fact that anti-aligned mosaics sample more unique spatial positions, and the result is that high output noise levels lead to anti-alignment under efficient coding.

Conversely, though not analyzed in [14], the observation that increasing input noise levels favor alignment can be explained through the same formalism. Whereas [14] found that, in the independent neuron case, increasing output noise drove increases in response threshold (to better encode stimuli above the noise floor), the same analysis shows that increases in input noise *decrease* response thresholds (Supplementary Figure 3). Moreover, [14] showed that the coactivation probability between an ON-OFF RF pair at a distance of $j$ lattice positions away in $d = 1$ with Gaussian inputs is given by

$$p_2(\tilde{\theta}, \rho_j) = \int_{\tilde{\theta}}^{\infty} \frac{1}{\sqrt{2\pi}} e^{-\frac{y^2}{2}} \left(1 - \Phi\left(\frac{\tilde{\theta} - \rho_j y}{\sqrt{1 - \rho_j^2}}\right)\right) dy, \tag{34}$$

where, for $g_j = \mathbf{w}_0^\top \mathbf{C}_x \mathbf{w}_j$ and $f_j = \mathbf{w}_0^\top \mathbf{w}_j$,

$$\tilde{\theta} = \frac{\theta}{\sqrt{g_0 + \sigma_{\text{in}}^2}}, \qquad \rho_j = \frac{g_j + \sigma_{\text{in}}^2 f_j}{g_0 + \sigma_{\text{in}}^2}$$

are the signal-to-noise-normalized threshold and unthresholded ON-OFF correlation, respectively. From this, it is clear (and Supplementary Figure 2A shows) that increasing $\sigma_{\text{in}}$ reduces $\tilde{\theta}$ not only through reductions in the optimal $\theta$, but through increases in the normalization factor. This, in turn, reduces $p_2$ through an increase in the integration limits in (34). Thus, increasing $\sigma_{\text{in}}$ results in a *narrower* "zone of independence" (Supplementary Figure 2B) around each RF, with the result that mosaics cannot anti-align without losing information.

## C    Sinusoidal temporal kernels in the absence of parameterization

Supplementary Figure 4 shows the distribution of temporal filters learned without using the parameterization in Equation 6, which have symmetric, sinusoidal shapes resembling Fourier bases. They also exhibit distinct clusters of temporal kernel shapes with varying spectral characteristics,

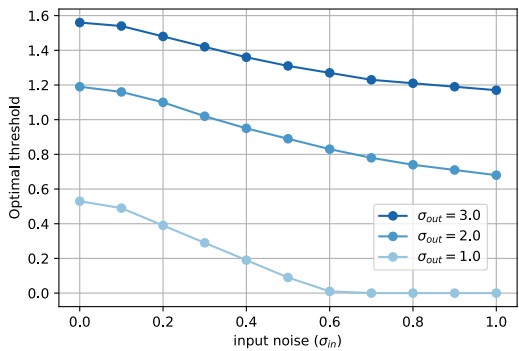

Supplementary Figure 3: Optimal thresholds decrease with input noise as they increase with output noise. Calculations based on the independent neuron model of [14].

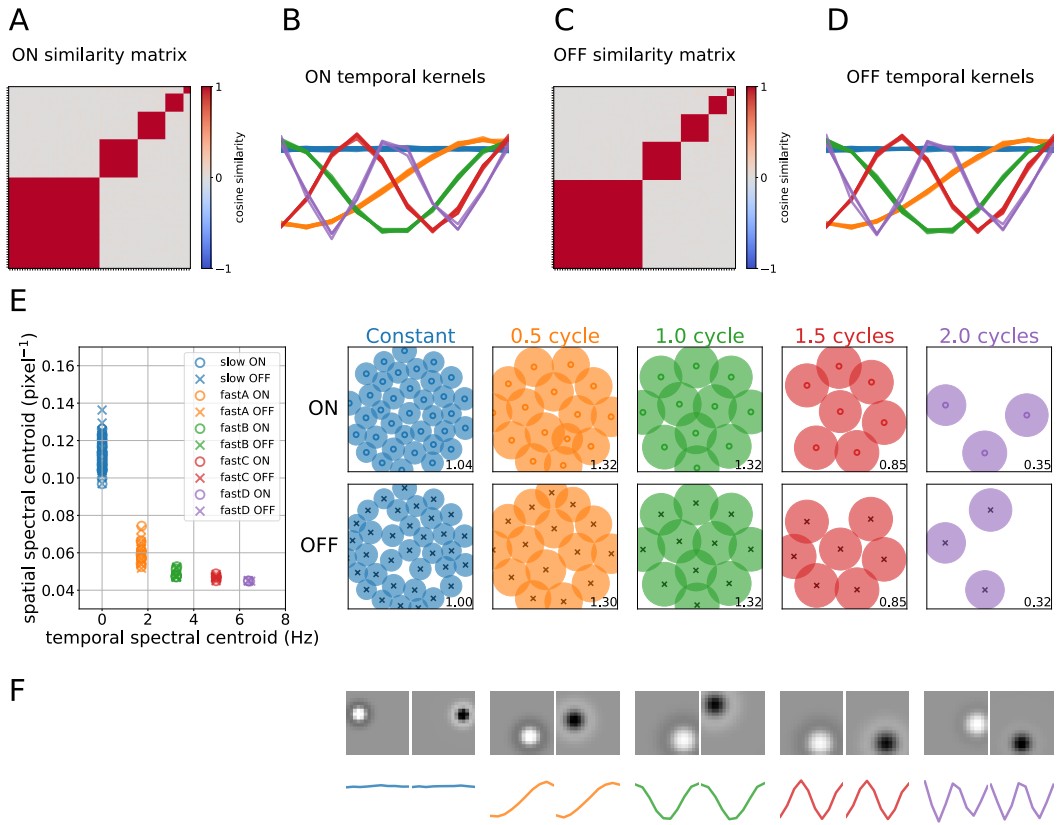

Supplementary Figure 4: Sinusoidal bases are learned when any unit-norm temporal kernels of a given length are allowed instead of using the parameterization in Equation 6. A model with $J = 144$ is used. (**A, C**) Self-similarity matrices of ON (A) and OFF (C) temporal filters show five distinct clutsers in each, with almost perfect orthogonality across clusters. The filters are sorted according to their spectral centroids. (**B, D**) The shapes of learned temporal ON (B) and OFF (D) filters, color-coded according to the clusters. Each cluster exhibits a sinusoidal shape with a distinct frequency, ranging from the constant to having roughly 0.5, 1.0, 1.5, and 2.0 cycles in the given kernel size. (**E**) (left) The distribution of the temporal and spatial spectral centroids. (right) ON and OFF mosaics corresponding to each RF type. The number in the corner of each plot denotes the coverage factor of the mosaic. (**F**) Learned shapes of a spatial ON and OFF filters (top) and the corresponding temporal filters (bottom) from each RF type.

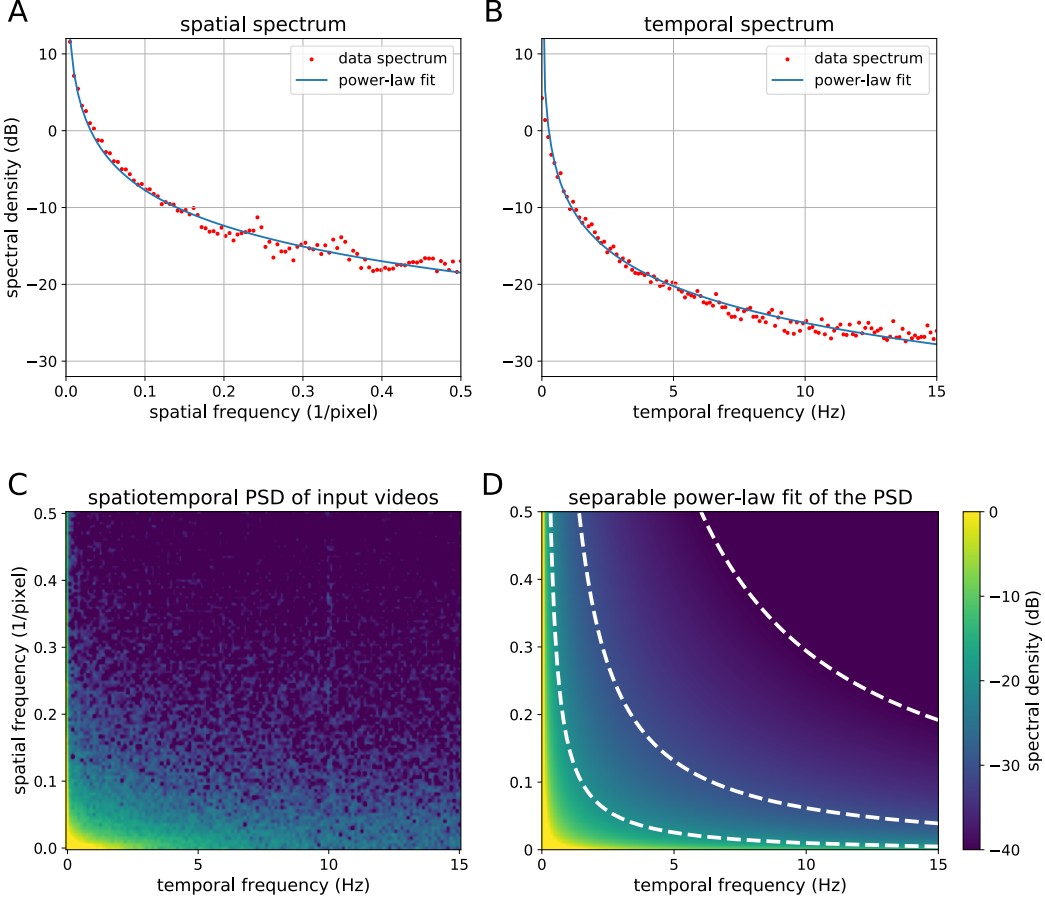

Supplementary Figure 5: The spatiotemporal spectrum of the input videos from the Chicago Motion Database is well-approximated by separable power-law fits of the spatial and temporal spectra. **(A)** The spatial spectrum of the data and its power-law fit. **(B)** The temporal spectrum of the data and its power-law fit. **(C)** The spatiotemporal power-spectral density of the data. **(D)** Reconstructed power-spectral density from the power-law fits. Contour lines indicate -40 dB, -30 dB, and -20 dB.

and each group corresponds to a spatial RF mosaic that tiles the space (Supplementary Figure 4E). Supplementary Figure 4A-D shows that the temporal filters are almost perfectly orthogonal across the groups, which allows independent information to be conveyed in each group. The shapes of the temporal filters obtained in the full model (Figure 3 and 4) can be interpreted as being as independent as possible while retaining biologically plausible filter shapes. In the case of auditory filters, it has been shown that efficient coding results in Fourier-like filters [27].

## D    Information gain upon adding individual RF mosaics

We examine how each of the RF mosaics contribute to the overall mutual information, by manually constructing four pairs of mosaics of spatial kernels placed in hexagonal grids, corresponding to four types of temporal filters: *Slow*, *FastA*, *FastB*, and *FastC* (6A). Given the 8 groups of spatiotemporal kernels, Figure 6B shows the order of adding mosaics such that each step brings the largest increase in the estimated mutual information. The optimal ordering indicates that it is always beneficial to add the slow mosaics first, in either order between ON and OFF (steps 1 and 2). Afterwards, the highest increase in MI is when adding one mosaic, either ON or OFF, from each shape in the order of increasing speed (steps 3-5), and then filling in the remaining mosaics again in the order of increasing speed (steps 6-8).

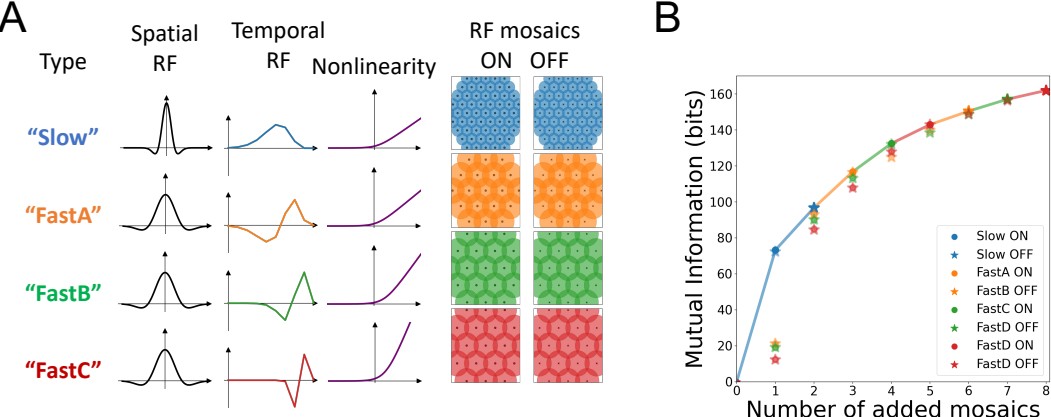

Supplementary Figure 6: The order of mosaic emergence is explained by the information gain upon adding individual mosaics. **(A)** Four sets of parameters are manually selected to construct spatial and temporal RFs, nonlinearities, and RF mosaics that resemble the *Slow*, *FastA*, *FastB*, and *FastC* RGCs. The mosaics are constructed to pack the space using a hexagonal pattern, resulting in $J = 234$ RGCs across the 8 mosaics. **(B)** Starting from an empty set, we repeat finding and adding the mosaic that maximally increases the mutual information, until all 8 mosaics are included. At each step, a scatter plot of the mutual information for each choice is plotted, and the solid line indicates the increase in mutual information by adding the best mosaic at each step.

## E    Additional Technical Details

**Determining the polarity of spatiotemporal kernels**    We used the parameterization of temporal kernels in Equation 6, and we additionally experimented with temporal kernels that can learn any unit-norm function in Supplementary Figure 4. Unlike that of spatial kernels in Equation 8 which has a positive peak intensity at the center, both parameterizations for temporal kernels are symmetric as to the positive and negative values the kernels can take, and there is no built-in indicator to distinguish ON and OFF kernels. Since ON and OFF RFs are the ones responding, respectively, to light increment and decrement [28], we determine the polarity of given temporal kernel $w[t]$ based on the sign of its cumulative response to the unit step function $u[t]$ up to the half duration of the kernel:

$$\sum_{t=0}^{\lfloor T/2 \rfloor - 1} (w * u)[t] = \sum_{t=0}^{\lfloor T/2 \rfloor - 1} \sum_{s=0}^{t} w[s] = \sum_{t=0}^{\lfloor T/2 \rfloor - 1} \left( \left\lfloor \frac{T}{2} \right\rfloor - t \right) w[t]. \tag{35}$$

In the visualizations of the spatial and temporal kernels, we flip the signs of the OFF temporal kernels as well as the corresponding spatial kernels, which keeps the separable spatiotemporal kernels unchanged (Equation 5). This makes the temporal kernels always visualized as ON filters, and the spatial kernels are displayed with a bright center for ON RFs and a dark center for OFF RFs. We also note that we plot the temporal kernels so that $t = 0$ is at the right and $T = t - 1$ is at the left of the graphs, making it comparable to the usual direction of visualizing temporal RFs, e.g. in spike-triggered averages.

**Data preprocessing and training**    The Chicago Motion Database [22] contains 257 natural video clips ranging from bees flying around the hive to waves approaching the shore. All video clips in the dataset are square-shaped and have varying sample rates. We resize and resample the video clips to 512×512 pixels and 30 fps using `ffmpeg`, in addition to converting it to grayscale by taking the luminosity channel of each frame using the Python Imaging Library. Following [10] and [14], we normalize each video to have zero mean and unit variance over all video dimensions, while the sliced video patches may have different means and variances. We used square-shaped input patches $\mathbf{x}(t)$ of size $D = 18^2 = 324$ pixels unless otherwise specified, and we applied a circular mask on the input patches following [14], which effectively constrained the spatial kernels within a circle bounded by the input square. Input patches consisted of either $T = 10$ or 20 frames sliced from 30-fps videos, and for computational efficiency, we used "valid"-mode convolutions with temporal kernels of the same size $T$, so each RGC yielded a single firing rate value $r_j$ for each input. Models

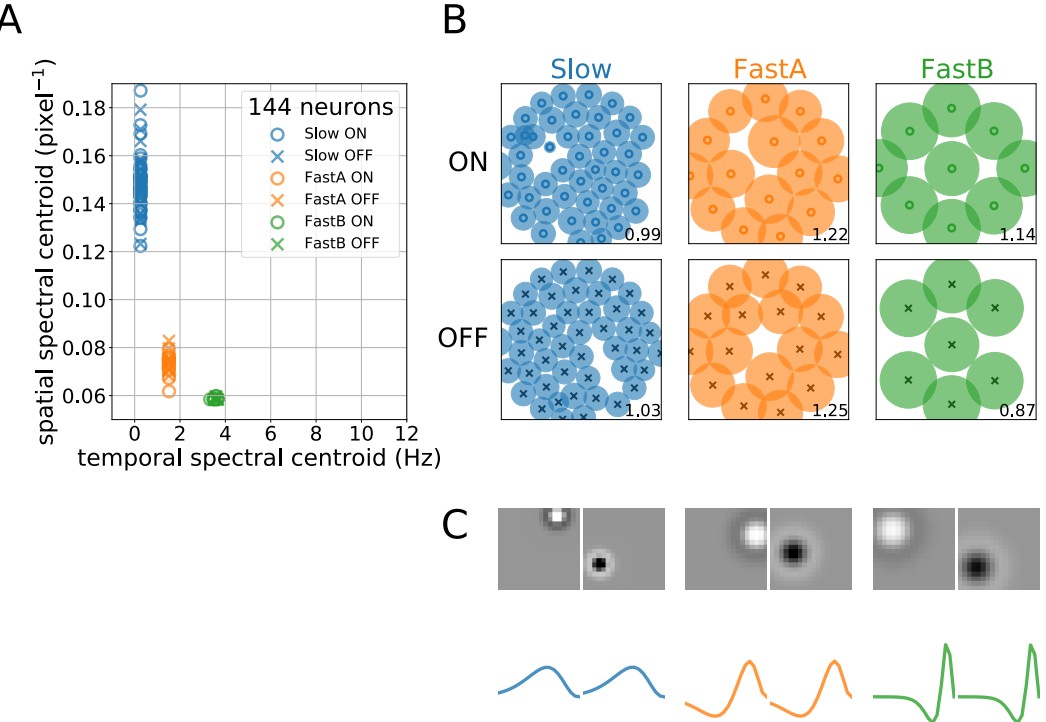

Supplementary Figure 7: Emergence of RF mosaics is not contingent on the choice of the hyperparameter $n$ in Equation 6. The above is the result of optimizing $J = 144$ RGCs using $n = 3$ instead of the default $n = 6$, showing the resulting distribution of spatiotemporal spectral centroids (A) and RF mosaics (B), as well as their spatial and temporal shapes (C). While the exact trends are different from the case of $n = 6$, this result indicates that the gradual emergence of distinct RF mosaics with increasing $J$ would happen with different values of $n$.

using larger convolutions, capturing information across both RGCs and time, produced similar results. For computational efficiency in large-scale experiments in Sections 4.1 and 4.2, we approximated the input data distribution as a multivariate Gaussian matched to the statistics of the natural image dataset estimated using one million samples of video patches. A batch of 128 such video patches was used for each update, and we used Adam optimizer [23] with a constant learning rate of 0.001. In Section 4.1, a model optimizing $J$ RGCs is trained for $5000J$ iterations, and each model in Section 4.2 is trained for 200,000 iterations. The experiments in 4.2 often result in either of aligned and anti-aligned configurations depending on the initialization of the model, so we repeat each configuration 10 times and select the result that obtained the highest mutual information objective (Equation 2). Each experiment in Sections 4.1, and 4.2 were run on a NVIDIA 1080 Ti GPU for about 50 hours, 20 hours, and 5 hours, respectively.

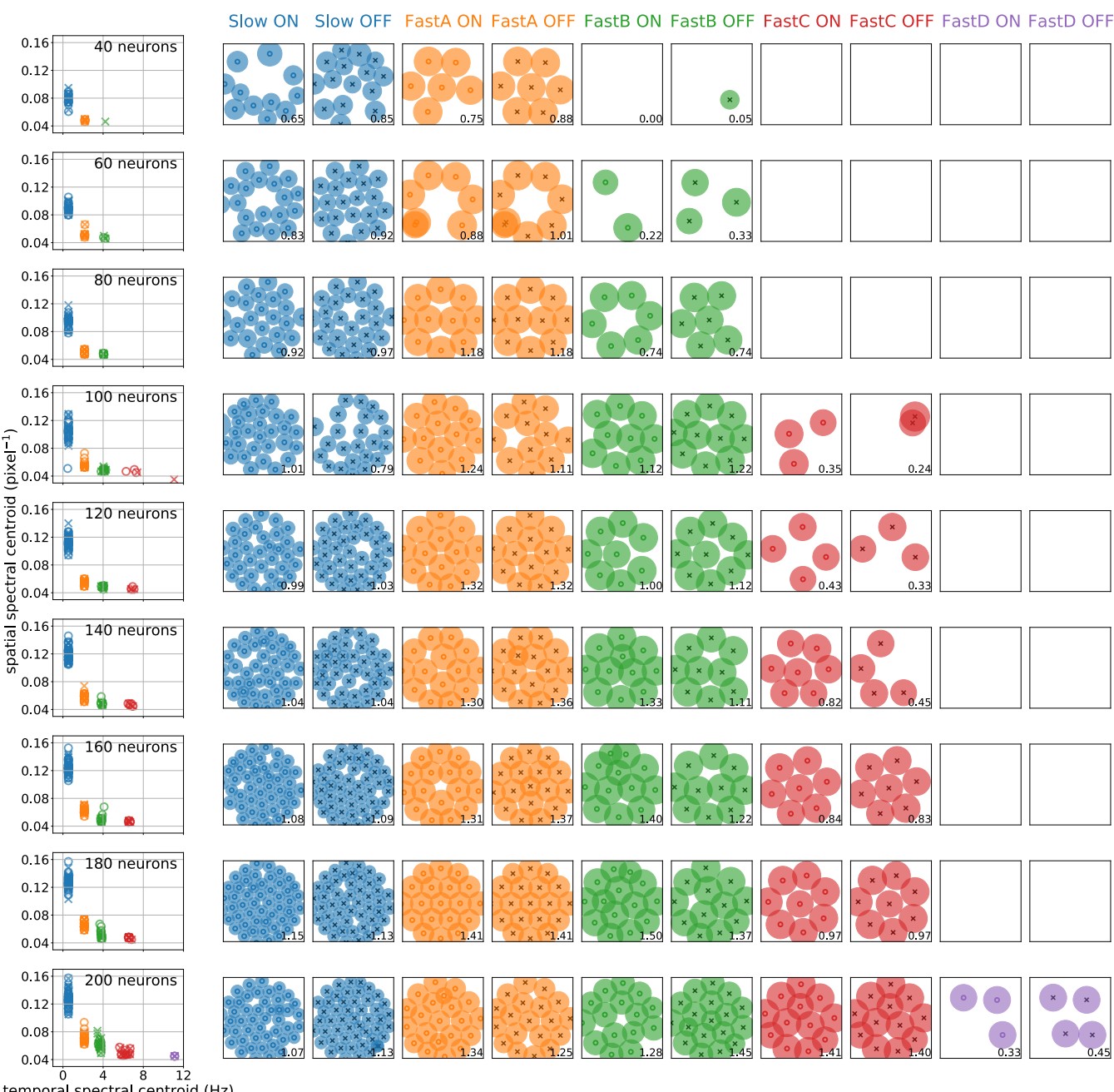

Supplementary Figure 8: The distribution of the spatial and temporal centroids of kernels (left), the RF mosaics (middle), and the temporal kernel shapes (right), as the number of RGCs under optimization increases from 40 to 200, by an increment of 20. As the number of RGCs increases, faster mosaics emerge to cover the higher-frequency region of the spatiotemporal spectrum, while the individual spatial RFs become more precise, using more number of RGCs to tile the same visual space.

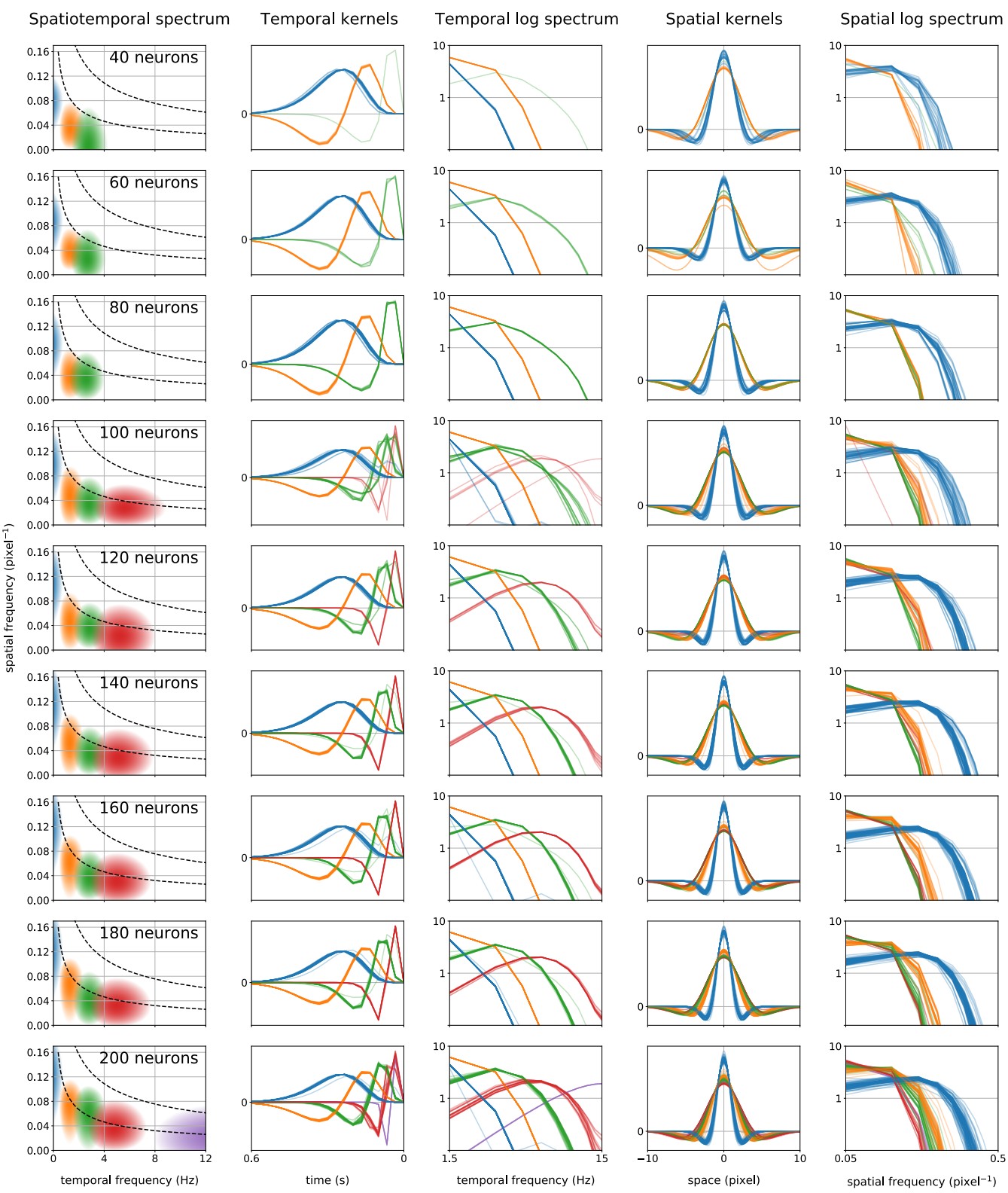

Supplementary Figure 9: More visualizations of the kernels in Supplementary Figure 8. Column 1 plots the region where the average spatiotemporal kernel of each type has high frequency response, superimposed with contour lines of the dataset's power spectral density. Each kernel type covers a different region of spatiotemporal frequencies, together packing increasingly larger areas where the spectral power of the dataset is concentrated. Individual temporal kernels are plotted in the time and frequency domains (columns 2-3), and spatial kernels in the space and frequency domains (columns 4-5).

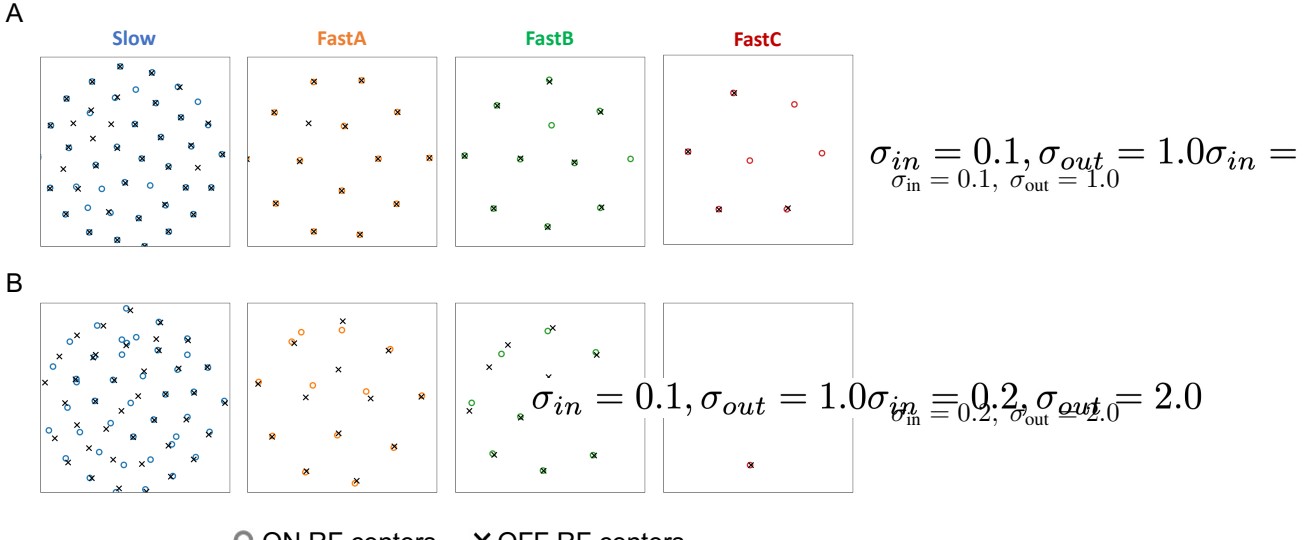

Supplementary Figure 10: RF centers per mosaic type under different input and output noise levels, using $J = 140$ RGCs. ON and OFF RF centers are more aligned under a lower noise level than under a higher noise level.