# OpenReview forum: "Efficient coding, channel capacity, and the emergence of retinal mosaics"
_NeurIPS.cc/2022/Conference — NeurIPS 2022 Accept_

### Official Review · Reviewer_QXXa · 2022-07-01

**Rating:** 8
**Confidence:** 4
**Soundness:** 3 good
**Presentation:** 3 good
**Contribution:** 4 excellent

**Summary:**

The authors study how retinal mosaics emerge from an efficient coding framework. They extend the framework to spatial-temporal kernels and video inputs, which is a crucial extensions to understand how natural environmental statistics influence coding in neural systems.  Also, the approach is highly innovative as it allows to derive multiple optimized mosaics.

**Questions:**

* Lines 124: The Bravais lattice is not sufficiently explained for non-physicists. The section up to Line 129 could be better explained. What are principal and reciprocal vectors?

* Fig 2A: The colored points should be defined in the legend.

* Fig. 5 is very dense and hard to understand - maybe more intuitive labels would help?




**Limitations:**

The authors should discuss the limitation of the efficient coding objective - is that really all that the organism cares about?

**Strengths And Weaknesses:**

Strengths:
* Extend previous models to spatiotemporal signals
* Study relationship between natural video statistics and optimal neural codes
* Interesting approach to derive multiple mosaics with interesting results

Weaknesses:
* Sometimes a bit jargony/dense - the authors should check how understandable each section is and make sure they can be followed by non-theoreticians
* Limited by focus on efficient coding objective - the authors could discucss whether reconstruction of the visual environment is really the ultimate goal of the visual system

---

> ### Author Response · Authors · 2022-08-02
> **addressing weaknesses and questions**
>
> ## Weaknesses:
> > Sometimes a bit jargony/dense - the authors should check how understandable each section is and make sure they can be followed by non-theoreticians
>
> Thank you. In our revised submission, we have particularly attempted to rework and clarify Section 3, which also caused confusion for some reviewers. We have reworked parts of the Introduction and Discussion to help clarify the main points.
>
> > Limited by focus on efficient coding objective - the authors could discucss whether reconstruction of the visual environment is really the ultimate goal of the visual system
>
> We agree that the efficient coding hypothesis is a strong one, and in a camera-ready version, we would like to discuss alternatives and related proposals in the discussion. However, we will also point out that our efficient coding objective only maximizes encoded information, without assuming that this information is subsequently used for reconstruction (unlike in Ocko et al. [10]). That is, our approach is agnostic about the ends for which information is encoded.
>
> ## Questions:
> > Lines 124: The Bravais lattice is not sufficiently explained for non-physicists. The section up to Line 129 could be better explained. What are principal and reciprocal vectors?
>
> We apologize for the confusion. In our revised manuscript, some technical details here have been moved to the appendices in the interest of a more streamlined presentation. Please also see our response to Reviewer 8e9N on this question.
>
> > Fig 2A: The colored points should be defined in the legend.
>
> Thank you. We have fixed this.
>
> > Fig. 5 is very dense and hard to understand - maybe more intuitive labels would help?
>
> Thank you for the suggestion. Our revised manuscript contains a new Figure 5 with some visual elements removed and enlarged for clarity, as well as more intuitive labels.

---

> > ### Comment · Reviewer_QXXa · 2022-08-02
> > **Thanks**
> >
> > Thank you for the clarification. I believe you work is a nice addition to the literature trying to understand the retina from a theoretical point of view.

---

> ### Author Response · Authors · 2022-08-02
> **Prefatory note**
>
> We have uploaded a revised submission in which we have attempted to address several questions shared by multiple reviewers. In particular, we have thoroughly revised Section 3, clarifying the analytical model and its relationship to the experiments in Section 4. We have also revised the introduction and discussion, where we clarify links to retinal physiology and the unique contributions of this work relative to Ocko et al. Since we are still restricted here to the nine pages of the initial submission, not the ten pages of the camera-ready version. Thus, some suggestions will be more fully addressed in the final preprint, but can only be partially addressed in this revision.

---

### Official Review · Reviewer_gDAm · 2022-07-06

**Rating:** 8
**Confidence:** 4
**Soundness:** 4 excellent
**Presentation:** 4 excellent
**Contribution:** 4 excellent

**Summary:**

The authors present a unified perspective on the appearance of different ganglion cell types (in terms of receptive field size, temporal properties and polarity) in the framework of efficient coding. They deploy a previously developed model and efficient coding framework to spatio-temporal movies. They show, first analytically in the case of a simplified linear model, later experimentally in the non-linear case, how optimal spatial and temporal filters change as the number of neurons (‘channel capacity’) increases. Additionally they investigate the resulting mosaics of similar types for (anti-) alignment and show that input as well as output noise has an important but opposed influence on the alignment of ON/OFF receptive fields.

**Questions:**

1. Fig. 2 a: How do the ‘experimental dots’ relate to the presented isoclines?
2. Could the authors comment on how their argumentation in line l. 186 (and also later) is depend on the assumption of time-space separability? While a detailed analysis is beyond the scope of the manuscript, the specialization to different regions of spatial and temporal frequencies is certainly influenced by this assumption.
3. l. 196: The data distribution was approximated by a multivariate Gaussian. Is this an appropriate approximations given the known power laws for natural scene statistics? How would a different distribution change the results?

**Limitations:**

The authors did not comment on limitations or negative societal impact. While latter remains quite abstract, the limitations could be indeed discussed in more detail.

**Strengths And Weaknesses:**

### Strengths

The authors embed their work excellently into the previous literature and work clearly out the parallels and their novel contributions. The theoretical as well as the experimental approaches are well described and cover different interesting arrangements and detailed analyses. All experiments are carried out carefully and are of very high quality, as well as the figures. The authors push the understanding of retinal layout further and try to unify previous approaches.

However, there are some minor weaknesses:

### Weaknesses
1. There are some open questions (see below).
2. A simple schema for the model would help to understand the setup.
3. From equation (2) it does not become clear what the differences to [7] are, and how the formula is derived (as I could not find the exact formulation in [7]).
4. Some parts of section 3 (and especially Suppl. A) are quite technical and it is sometimes difficult to follow. The authors could try to strengthen a ‘red thread’ and give also an intuition where possible.
5. Sections 3 and 4 seem to be rather detached from each other. While formulas are derived thoroughly, their interpretation and link to section 4 could be improved.
6. Their final conclusion that previous cell types change down in temporal frequency (l. 274) is not well supported by the presented experiments. While it is more obvious for the spatial frequency, an additional analysis for the temporal domain could help to support this argument.


### Minor comments:
- l. 256: shouldn’t it reference to eq (8)?

---

> ### Author Response · Authors · 2022-08-02
> **Overall evaluation and weaknesses**
>
>  > All experiments are carried out carefully and are of very high quality, as well as the figures. The authors push the understanding of retinal layout further and try to unify previous approaches.
>
> We appreciate this comment, since this work replicates, unifies, and extends previous results of Atick and Redlich (1990, 1992), Karklin and Simoncelli (2011), Ocko et al. (2018), and Jun et al. (2021) in a single coherent framework.
>
> ## Weaknesses
> > 2. A simple schema for the model would help to understand the setup.
> We have now updated Figure 1 in our revised submission to include an additional model schematic (new panel A) showing the roles of input noise, output noise, filters, and nonlinearities in the transition from input images to firing rates.
>
> > 3. From equation (2) it does not become clear what the differences to [7] are, and how the formula is derived (as I could not find the exact formulation in [7]).
>
> Our form of the objective function follows Jun et al. [11], who derived it from [7] using the Matrix Inversion Lemma. We have clarified this in the text.
>
>
> > 4. Some parts of section 3 (and especially Suppl. A) are quite technical and it is sometimes difficult to follow. The authors could try to strengthen a ‘red thread’ and give also an intuition where possible.
>
> We appreciate this suggestion and have attempted in our rewritten Section 3 in the revised manuscript to do precisely this. We have likewise substantially expanded Appendix A.5 to give a more thorough exposition of these results.
>
>
> > 5. Sections 3 and 4 seem to be rather detached from each other. While formulas are derived thoroughly, their interpretation and link to section 4 could be improved.
>
> We agree. In our rewritten Section 3, we have tried to make clearer the key analytical results that bolster our simulations in Section 4.
>
> > 6. Their final conclusion that previous cell types change down in temporal frequency (l. 274) is not well supported by the presented experiments. While it is more obvious for the spatial frequency, an additional analysis for the temporal domain could help to support this argument.
>
> The reviewer is correct. This was a mistake on our part, and we have removed this claim.

---

> > ### Comment · Reviewer_gDAm · 2022-08-05
> > **Re**
> >
> > I thank the authors for their detailed response and clarifications.
> >
> > At this point, I have no further questions.
> >
> > I think it is a very strong and thoroughly worked-out submission.

---

> ### Author Response · Authors · 2022-08-02
> **Questions**
>
>
> > 1. Fig. 2 a: How do the ‘experimental dots’ relate to the presented isoclines?
>
> Figure 2A in the original submission erroneously plotted the isoclines for exponential, not power-law spectra. In our revised manuscript, Figure 2A plots the correct power law isoclines, and the dots roughly follow one of these.
>
> > 2. Could the authors comment on how their argumentation in line l. 186 (and also later) is depend on the assumption of time-space separability? While a detailed analysis is beyond the scope of the manuscript, the specialization to different regions of spatial and temporal frequencies is certainly influenced by this assumption.
>
> This is an excellent question. In Appendix A.3, we derive the optimal spacetime filter for generic $C_x(k, \omega)$ and find that the result is not, in general, space-time separable. In fact, the optimal filter need not be separable even when $C_x$ is (cf. ll. 541-551 in A.4). However, the argument for spectrally non-overlapping filters does not depend on separability, either of the filter or the input image spectrum. That is, $v’(k, \omega)v^*(k, \omega) = 0$ is always sufficient (we do not prove necessity) to render information from the different mosaics additive, since that removes all cross-terms between mosaics in the determinants of (2).
>
> > 3. l. 196: The data distribution was approximated by a multivariate Gaussian. Is this an appropriate approximation given the known power laws for natural scene statistics? How would a different distribution change the results?
>
> This is an excellent question. Three points:
> 1. While our linear analysis considered only Gaussian statistics (with power law correlations), our simulations in Section 4 used natural videos (including many higher-order moments). For this reason, we do not believe that non-Gaussianity is essential to our results.
>
> 2. However, for light-tailed statistics (e.g., exponential tails), we do expect differences. For instance, while Atick and Redlich considered exponential correlation functions in their original treatment of the spatial filters and found similar shapes to those in Figure 2B, Jun et al (reference 14 in the manuscript) found that center-surround receptive fields failed to form for Gaussian images in the model of Karklin and Simoncelli, whether or not tails were power-law (reported in the supplementary material of that work).
>
> 3. Finally, for exponential correlations, all information is local, and thus receptive fields should be somewhat smaller and fail to “repel” one another unless they overlap. This should result in mosaics that are more “fluid-like,” with only short-range interactions and without long-range mosaic order.
>
> In summary, while higher-order correlations (i.e., non-Gaussianity) do not appear necessary to drive the effects we see, the existence of power-law correlations in images appears to be critical.

---

> ### Author Response · Authors · 2022-08-02
> **Prefatory note**
>
> We have uploaded a revised submission in which we have attempted to address several questions shared by multiple reviewers. In particular, we have thoroughly revised Section 3, clarifying the analytical model and its relationship to the experiments in Section 4. We have also revised the introduction and discussion, where we clarify links to retinal physiology and the unique contributions of this work relative to Ocko et al. Since we are still restricted here to the nine pages of the initial submission, not the ten pages of the camera-ready version. Thus, some suggestions will be more fully addressed in the final preprint, but can only be partially addressed in this revision.

---

### Official Review · Reviewer_kZcp · 2022-07-11

**Rating:** 7
**Confidence:** 3
**Soundness:** 3 good
**Presentation:** 2 fair
**Contribution:** 2 fair

**Summary:**

This paper presents a model for retinal mosaic organization, derived
by application of the efficient coding principle to natural movies. In
particular, the model shows that the total number of retinal ganglion
cells is a key parameter that controls the emergence of distinct cell
type. The model is also used to extract predictions about the relative
phase of distinct retinal mosaics as a function of input noise.


**Questions:**

1. Is it possible to clarify better how the differences between this
   work and [10] (listed in the "related work" paragraph) affect the
   main conclusions of the paper? Or is the main point that the
   results here are qualitatively similar to those in [10], despite
   the differences listed under "related work"? (I am referring mostly
   to the material in Section 3). For instance, the first sentence of
   the Discussion reads: *"Here, we have shown that efficient coding of
   natural movies results in the formation of multiple receptive field
   mosaics with complementary filtering properties [10]."* Is this
   suggesting that the main result of the paper is something for which
   ref [10] should be credited?
2. Lines 50-53 read: *"In the case of linear encoding, we show
   analytically how a tradeoff between information gains from
   increasing mosaic density and informationg gains from adding new,
   specialized cell types leads to the addition of new mosaics
   capturing higher temporal frequency information."* Can you make sure
   that the results backing up this statement are easier to find in
   the text? If I am not mistaken, the passage should be lines
   182--189. That part of the text hinges on the fact that *"as
   detailed above, adding filters covering non-overlapping spectral
   windows allows mosaics to specialize to different regions of
   spatial and temporal frequency"*. The "above" here probably refers
   to the previous paragraph, that ends with the following: *"In [10],
   the authors considered a form of this with spatial bandwidth
   partitioning (v(k)\*v'(k) = 0), but as we show below, this special
   case encounters a limit beyond the first two mosaics, when the
   spatial passband spectra of new filters substantially
   overlap. Rather, the more general solution is that mosaic filters
   are band-limited in both space and time, arranging themselves to
   tile minimally overlapping regions with highest power in the (k,ω)
   plane (Appendix A)."* This passage, in turn, contains a reference to
   ref [10], one to "below", and one to the Appendix. Forgetting about
   [10] and assuming that "below" here refers to Section 4 (which is
   not valid for the original point we were trying to find support
   for, as we're looking for an analytical solution, as claimed in the
   Introduction), we therefore follow this chain of references to the
   Appendix. The relevant section seems to be section A.4, "Effects of
   channel capacity and new mosaic formation", but it would be great
   to have this spelled out more clearly, and perhaps the result in
   the Appendix also connected back to the claim in the main text in a
   more explicit way.
3. The Discussion states (lines 278--279) that the results are "in
   strong agreement with observed retinal data". I found this
   statement surprising, given the absence of quantitative comparisons
   with empirical data in the paper. Is it possible to clarify what is
   meant by this, and to substantiate the statement with direct
   comparisons?

**Limitations:**

The limitations of the work are acknowledged very briefly in the
discussion (*"despite the strong assumptions of our model - linear
filtering, separable filters, firing rates instead of spikes -"*), but
not unsuitably so for a modelling work of this type.


**Strengths And Weaknesses:**

### Strengths
The paper tackles a relevant problem, that is modeling the
emergence of distinct retinal mosaics from an efficient coding
perspective, by building on a solid foundation (reference [10]).

### Weaknesses
The paper is very hard to read, mostly because (1) its
tight coupling with reference [10], and (2) how much of it is
relegated to the Appendix. With respect to (1), I am not sure I
understand fully what parts of the model, and which results, are fully
novel here and which are minor refinements or variants of the results
in [10]. Regarding (2), the Appendix includes not only derivations for
most of the equations shown in the main text, or technical details for
the realization of some of the plots, but also - to my understanding -
the basic logic of some of the results (I'll give more details on this
below, under "questions").

Additionally, the paper claims "strong agreement with observed data" but contains no data and no quantitative assessment of such claim.

---

> ### Author Response · Authors · 2022-08-02
> **readability and our contribution**
>
> > The paper tackles a relevant problem, that is modeling the emergence of distinct retinal mosaics from an efficient coding perspective, by building on a solid foundation (reference [10])
>
> > The paper is very hard to read, mostly because (1) its tight coupling with reference [10], and (2) how much of it is relegated to the Appendix. With respect to (1), I am not sure I understand fully what parts of the model, and which results, are fully novel here and which are minor refinements or variants of the results in [10].
>
> We apologize for the dense text: upon re-reading we agree there was substantial room for improvement. While space constraints force us to limit most technical details to the appendix, we agree that section 3 was muddled and have thoroughly rewritten it in the revised manuscript. We have also substantially expanded and clarified our treatment of new mosaic formation in Appendix A.5, and Figure 2 contains three additional panels illustrating the effect of changing mosaic density on receptive field size, the marginal information gain per neuron, and the allocation of RFs to each mosaic as new neurons are added.
>
> Moreover, the question of the relation between this work and [10] (Ocko et al.) is an important one, which we have attempted to better address in the Related Work section of our revised Discussion and briefly recapitulate here:
> - [10] addressed the question of encoding benefits of multiple cell types in the retina, finding that four types (ON and OFF cells for two filters) had benefits over two types (ON and OFF for a single filter) when considering the problem of reconstructing visual inputs from retinal activity.
> - To do this, [10] used a setup with a fixed number of cell types in fixed mosaics in a convolutional neural network and optimized both filter shapes and cell spacing using Gaussian white noise.
> - By contrast, our work takes an efficient coding approach (maximizing information encoded, not reconstruction) and answers the questions “How many cell types should the retina use?” and “How does this vary as a function of neuron number?”
> - Our work does not a priori assume a particular number of cell types, mosaic configurations, or shared filter shapes. Instead, these emerge from the optimization itself. Furthermore, we optimize using natural videos, not just Gaussian noise. In addition, our approach, as pointed out by reviewer dDAm, unifies several approaches in the literature, including those of Karklin & Simoncelli (2011) with Atick and Redlich (1990).
> - Finally, our results on mosaic alignment extend the work of Jun et al. (2021) to the spacetime case, a problem not previously solved.  Optimizing a model of efficient coding in space and time together is novel to our knowledge.
>
> In addition, it is important to note that while we have attempted to note points of similarity between [10] and this work, our mathematical formulation is distinct and our presentation entirely self-contained.

---

> > ### Comment · Reviewer_kZcp · 2022-08-07
> > **Response to authors**
> >
> > Thank you for your efforts in addressing the points I raised in my review, particularly the relationship with Ocko et al, the reference to specific empirical observations that are reproduced by the model, and the reorganization of Section 3. I think these changes have made the paper more accessible, and I will update my score accordingly.

---

> ### Author Response · Authors · 2022-08-02
> **replies to other questions**
>
> > Regarding (2), the Appendix includes not only derivations for most of the equations shown in the main text, or technical details for the realization of some of the plots, but also - to my understanding - the basic logic of some of the results (I'll give more details on this below, under "questions").
>
> We regret that space constraints did not permit us to incorporate more of these results into the main text. However, in our revised Section 3, we have attempted to provide a more self-contained presentation of the key results with more specific pointers to the Appendices.
>
> > Additionally, the paper claims "strong agreement with observed data" but contains no data and no quantitative assessment of such claim.
> We soften this statement and now explain it more (lines 258-262):
>
> “Furthermore, despite the assumptions of this model---linear filtering, separable filters, firing rates instead of spikes---our results are consistent with observed retinal data. For example, RGCs with small spatial RFs exhibit more prolonged temporal integration: they are also more low-pass in their temporal frequency tuning. Second, there is greater variability in the size and shape of spatial RFs at a given retinal location, but temporal RFs exhibit remarkably little variability in our simulations and in data \cite{ravi2018}. Thus, these results further testify to the power of efficient coding principles in providing a conceptual framework for understanding the nervous system.”
>
> > 1. Is it possible to clarify better how the differences between this work and [10] (listed in the "related work" paragraph) affect the main conclusions of the paper?
>
> > Or is the main point that the results here are qualitatively similar to those in [10], despite the differences listed under "related work"? (I am referring mostly to the material in Section 3). For instance, the first sentence of the Discussion reads: "Here, we have shown that efficient coding of natural movies results in the formation of multiple receptive field mosaics with complementary filtering properties [10]." Is this suggesting that the main result of the paper is something for which ref [10] should be credited?
>
> As detailed above, while we address the same questions, we believe this work represents a substantial advance in generality beyond the results presented in [10]. Our work - to our knowledge – is completely new and departs from [10] by answering the following question: how many RGC types should be to used to efficiently encode natural videos, and how does this vary with the number of cells available?
>
> > 2. Lines 50-53 read: "In the case of linear encoding, we show analytically how a tradeoff between information gains from increasing mosaic density and information gains from adding new, specialized cell types leads to the addition of new mosaics capturing higher temporal frequency information." Can you make sure that the results backing up this statement are easier to find in the text?
>
> We agree that this presentation was confusing and direct the reviewer to the rewritten Section 3 in the revised manuscript.
>
> > The Discussion states (lines 278--279) that the results are "in strong agreement with observed retinal data". I found this statement surprising, given the absence of quantitative comparisons with empirical data in the paper. Is it possible to clarify what is meant by this, and to substantiate the statement with direct comparisons?
>
> As mentioned above, we have softened this statement and now explain it more (lines 258-262).

---

> ### Author Response · Authors · 2022-08-02
> **Prefatory note**
>
> We have uploaded a revised submission in which we have attempted to address several questions shared by multiple reviewers. In particular, we have thoroughly revised Section 3, clarifying the analytical model and its relationship to the experiments in Section 4. We have also revised the introduction and discussion, where we clarify links to retinal physiology and the unique contributions of this work relative to Ocko et al. Since we are still restricted here to the nine pages of the initial submission, not the ten pages of the camera-ready version. Thus, some suggestions will be more fully addressed in the final preprint, but can only be partially addressed in this revision.

---

### Official Review · Reviewer_8e9N · 2022-07-16

**Rating:** 4
**Confidence:** 3
**Soundness:** 3 good
**Presentation:** 2 fair
**Contribution:** 3 good

**Summary:**

Presents mathematical analysis and computer simulation of model that maximizes mutual information between photoreceptors and the rgc outputs of the retina.  Shows that several different spatiotemporal filters are derived that mosaic the retina in different ways.

**Questions:**

see above

**Limitations:**

see above;  no societal impact issues.


**Strengths And Weaknesses:**

Strengths

A nice formulation and modeling approach to understand retinal function.  Impressive technical work.

Weakness

Not sure what the main take away is.  The goal appears to be to understand the neural encoding in the retina, but after that the analysis and results, there is no attempt to tie these back to neurobiological mechanisms.  It seems one could, but the paper just ends with the statement, "our results are in strong agreement with observed retinal data," which leaves you hanging.

Specific issues:

The difference of Gaussians model in eq. 8: it mentions that the center position of each kernel is different for each neuron, but is this also learned?  not mentioned.

Section 3:  linear model in the continuum limit - this is very unclear.  what is being continuized?  space?  The integral is over frequency space - not following what's going on.  principal vectors a_1, a_2 and reciprocal vectors b_1, b_2 - what are these?

Section 4.1: " power spectral density can be well approximated by a product of spatial and temporal power-law densities" -  Dong & Atick is cited, but curiously the claim the exact opposite, it is not separable.

Figure 4, panel A shows striking clustering in temporal spectral centroids - they are all stacked neatly in tight columns, no scatter.  is this what emerges from the learned filters, or is somehow the quantization imposed?

The mosaics are interesting to look at, but not clear what to take away from this.

Overall this seems like a very promising direction, I want to like this paper, but I find it a bit confusing and lacking a clear message.

---

> ### Author Response · Authors · 2022-08-02
> **Prefatory note:**
>
> We have uploaded a revised submission in which we have attempted to address several questions shared by multiple reviewers. In particular, we have thoroughly revised Section 3, clarifying the analytical model and its relationship to the experiments in Section 4. We have also revised the introduction and discussion, where we clarify links to retinal physiology and the unique contributions of this work relative to Ocko et al. Since we are still restricted here to the nine pages of the initial submission, not the ten pages of the camera-ready version. Thus, some suggestions will be more fully addressed in the final preprint, but can only be partially addressed in this revision.

---

> ### Author Response · Authors · 2022-08-02
> **main takeaway**
>
> > Not sure what the main takeaway is. The goal appears to be to understand the neural encoding in the retina, but after that the analysis and results, there is no attempt to tie these back to neurobiological mechanisms. It seems one could, but the paper just ends with the statement, "our results are in strong agreement with observed retinal data," which leaves you hanging.
>
> We apologize for insufficient clarity on this point.  The main takeaway is that the number of cell types produced by efficient coding theory depends on the channel capacity (number of RGCs that the system can optimize). When there are a small number of cells to optimize, the system produces 2-4 cell types, however, when there are lots of cells available, it can produce 8-10 cell types. To our knowledge, this is a completely novel observation about efficient coding that links it to cell type diversity. Furthermore, we develop the formalism (sketched in Section 3, fully detailed in Appendix A) to show when and why new cell types emerge as more cells are provided to the optimization. Finally, each new cell type that emerges in the optimization exhibits a progressively larger spatial receptive field and briefer (and more biphasic or bandpass) temporal integration. This matches how retinal ganglion cells sample space and time across a diverse array of species. While this last point is similar to previous work (Ocko et al), that study employed an optimization in which RF spacing and RF number were explicitly constrained to be inversely correlated, while in our study, the mosaic organization, cell densities, and number of cell types, emerge from the optimization (this structure is not imposed). We have added some brief text at the end of the manuscript to make these connections between the theory and retinal biology (lines 258-262), and we will expand on this in a camera-ready version.

---

> ### Author Response · Authors · 2022-08-02
> **answers to other questions**
>
> > The difference of Gaussians model in eq. 8: it mentions that the center position of each kernel is different for each neuron, but is this also learned? not mentioned.
>
> Correct, these are learned together with the other parameters mentioned in the section. We have clarified this in the revised manuscript.
>
> > Section 3: linear model in the continuum limit - this is very unclear. what is being continuized? space? The integral is over frequency space - not following what's going on. principal vectors a_1, a_2 and reciprocal vectors b_1, b_2 - what are these?
>
> We apologize for the confusion. The linear model is of an infinite retina, which leads to a continuum in frequency space. The title of this section has been changed to “Linear model in the infinite retina limit” in the revised manuscript to more accurately reflect this.
>
> The vector notation comes from crystallography, and was inadequately explained. The vectors $\mathbf{a}_1$ and $\mathbf{a}_2$ are basis vectors for the lattice formed by the retinal mosaic, such that $n_1 \mathbf{a}_1$ and $n_2 \mathbf{a}_2$ are the locations of receptive field centers for all integers $n_1$ and $n_2$. The reciprocal vectors are those vectors for which $\\mathbf{a}\_i \\cdot \\mathbf{b}\_j = 2\\pi \\delta\_{ij}$. That is, they are the frequency vectors with respect to which the system is periodic. The revised manuscript has removed this notation-heavy text and replaced it with a more intuitive definition and a fuller explanation of the notation in Appendix A.1.
>
> > Section 4.1: " power spectral density can be well approximated by a product of spatial and temporal power-law densities" - Dong & Atick is cited, but curiously they claim the exact opposite, it is not separable.
>
> Correct, Dong and Atick found that the spectrum was generally non-separable in space and time. However, in the same work, they derived a limit in which the spectrum became separable, taking the form $C_x(k, \omega) = A/k^\alpha\omega^2$. Ocko et al. used $\alpha=2$, whereas we use $\alpha \approx 1.3$, the value calculated by Dong and Atick. We also note that the factorized power spectrum is a good approximation to the spectrum we calculated in the natural video training set we used; we have added Supplementary Figure 4 visualizing the separable power-law fits.
>
> > Figure 4, panel A shows striking clustering in temporal spectral centroids - they are all stacked neatly in tight columns, no scatter. is this what emerges from the learned filters, or is somehow the quantization imposed?
>
> We did not impose this quantization, and there are small variations in the parameters determining the temporal filters, but they are less variable than those determining the spatial filters in each case, as Figure 4A shows. We suspect this is because, while both spatial and temporal filters are controlled by similar numbers of parameters, the number of frames (20) over which the temporal filters are defined is much smaller than the number of pixels over which the spatial filters are defined, so that there is more resolution afforded in the Fourier representation of the kernels in the spatial case. Note also that for larger numbers of neurons (e.g., the third row in 4A), the green and red subtypes exhibit more variation.
>
> > The mosaics are interesting to look at, but not clear what to take away from this..
>
> We would argue that plots of the mosaics make clear several features of mosaic formation noted in our revised Section 3 and more fully explained in Appendix A.5:
> - Receptive fields for each mosaic reduce in size as the number of neurons increases. This is reflected in the increase in spatial centroid of the learned RFs in the plots to the left as we look down the rows.
> - Consequently, mosaic density increases once mosaics are “full” (i.e., space is fully tiled).
> - Newly added neurons are unevenly distributed across mosaics. That is, new RFs continue to be added to the blue mosaic even as the red and purple mosaics are forming (cf. also the new Figure 2E).
> - When a new cell type emerges, the corresponding RFs do not completely fill the available space (e.g. FastB cells in top row of Figure 3B).
> In an expanded camera-ready version, we will use additional space in the caption and text to draw attention to these features.

---

### Meta-Review · Area_Chair_w9vB · 2022-08-26

**Recommendation:** Accept
**Confidence:** Certain

**Metareview:**

This paper received 1 accept, 2 strong accepts and 1 reject. All reviewers agree that the proposed model is elegant and that the technical work is impressive (even the negative reviewer). The main criticism of the negative reviewer is that  the main take away is not clear. The authors submitted a revised version of the manuscript. Sadly, the reviewer did not read the rebuttal and/or engage in a discussion post-rebuttal. The AC considers that the main criticism of this reviewer was addressed. In light of this, the AC recommends the paper to be accepted.

**Award:**

No

---

### Decision · Program_Chairs · 2022-09-14

Accept